# Multispecies deep learning using citizen science data produces more informative plant community models

Philipp Brun [1] ✉, Dirk N. Karger [1], Damaris Zurell [2], Patrice Descombes [3,4], Lucienne C. de Witte [1], Riccardo de Lutio [5], Jan Dirk Wegner [6] & Niklaus E. Zimmermann [1]

In the age of big data, scientific progress is fundamentally limited by our capacity to extract critical information. Here, we map fine-grained spatio-temporal distributions for thousands of species, using deep neural networks (DNNs) and ubiquitous citizen science data. Based on 6.7 M observations, we jointly model the distributions of 2477 plant species and species aggregates across Switzerland with an ensemble of DNNs built with different cost functions. We find that, compared to commonly-used approaches, multispecies DNNs predict species distributions and especially community composition more accurately. Moreover, their design allows investigation of understudied aspects of ecology. Including seasonal variations of observation probability explicitly allows approximating flowering phenology; reweighting predictions to mirror cover-abundance allows mapping potentially canopy-dominant tree species nationwide; and projecting DNNs into the future allows assessing how distributions, phenology, and dominance may change. Given their skill and their versatility, multispecies DNNs can refine our understanding of the distribution of plants and well-sampled taxa in general.

During the last decade, we have witnessed a shift away from traditional expert-led biodiversity surveys towards data gathered via fast-growing citizen science platforms[1,2]. This development has resulted in an exponential increase in observational data, providing detailed information not only on locations of species observations, but also on their temporal dynamics[3,4]. Such information is dearly needed to better understand biodiversity patterns and their underlying drivers, and to inform efficient conservation actions[5]. Yet, most citizen science observations are made opportunistically, which leads to significant spatial, temporal, and taxonomic biases[6] in the data. Spatial sampling biases are typically related to the accessibility of a location[7], while temporal sampling biases are linked to periods during which species are readily observed and/or easily identified (e.g. the flowering period).

Taxonomic reporting biases exist because citizen scientists prefer to report occurrences of conspicuous and charismatic species and taxa[8]. For analyzing biodiversity[9] these biases pose a major challenge, and therefore methods and approaches that can efficiently handle them are indispensable.

During recent decades, species distribution models (SDMs) have been among the most commonly used tools to model biodiversity from species observations[10]. SDMs are a host of algorithms used to characterize habitat suitability and potential distributions of individual species as a function of their environmental niches[11]. When fitting SDMs with opportunistic citizen science data, disentangling species' environmental preferences from spatial sampling bias is a key challenge[12]. Spatial sampling bias can be reduced by spatial or

[1]Swiss Federal Research Institute WSL, 8903 Birmensdorf, Switzerland. [2]Institute of Biochemistry and Biology, University of Potsdam, 14469 Potsdam, Germany. [3]Muséum cantonal des sciences naturelles, département de botanique, 1007 Lausanne, Switzerland. [4]Department of Ecology and Evolution, University of Lausanne, 1015 Lausanne, Switzerland. [5]EcoVision Lab, Photogrammetry and Remote Sensing, ETH Zurich, 8092 Zürich, Switzerland. [6]Department of Mathematical Modeling and Machine Learning, University of Zurich, 8057 Zurich, Switzerland. ✉e-mail: philipp.brun@wsl.ch

environmental thinning and pooling of observations[13], and by manipulating the background points (or pseudoabsences)[14,15]. Thinning observations, however, leads to a considerable loss of information, which is then lacking to investigate aspects like the seasonal turnover of biodiversity. Moreover, sophisticated procedures of spatial bias correction are computationally demanding when applied to many species[16].

Driven by their success in data science[17], deep neural networks (DNNs) have become an increasingly popular alternative to model biodiversity from observational data[18–24]. Compared to SDMs, DNN-based modeling frameworks offer interesting new perspectives. They allow considering the spatial configuration of a landscape as predictor[22], and they can cope with huge data sets[21,22]. Additionally, DNN-based approaches typically model the distributions of many species jointly, and multispecies models have been shown to be comparably robust to spatial sampling bias when drawing inference from presence-only data[25]. If spatial variations in sampling intensity are similarly represented in the occurrence records of a group of species, they have a smaller effect on the relative observation probability of one species versus another, than on habitat suitability scores derived from contrasting records of individual species against random background points[16]. Lower susceptibility to sampling bias, in turn, reduces the need to sacrifice observations through thinning and allows harnessing information from the full dataset, for example, the explicit consideration of seasonal effects on observed biodiversity[19,20]. Immediate ways to cope with taxonomic reporting bias, on the other hand, are not offered by DNNs.

DNNs use cost functions to quantify how well model predictions match with observations. A powerful cost function should be computationally efficient, numerically robust, and designed for the problem at hand. A commonly-used cost function that has been employed successfully in multispecies distribution modeling[22,26] is the cross-entropy loss[27] (CEL). It is efficient and robust, and has the desirable property of directing the DNN to learn true conditional probabilities[28]. Yet, the CEL assumes classes to be mutually exclusive, and thus exactly one species to be present per observation (although multiple observations may exist for the same spatiotemporal location). Presence-only observations, however, are single positive multi-label data[29], meaning that we expect multiple species to be present per observation, but only have information about the presence of one of them (and no information about the presence or absence of all other species). When training computer vision models with this type of data, cost functions accounting for incomplete information have been shown to be advantageous over the CEL[29]. One way to avoid expecting single-species presence is framing the learning task as a ranking problem, where the most relevant species identities are sought for a given set of environmental conditions. Such a framing is analogous to ranking the relevance of websites with respect to a given search query[30]. Although less common in ecology, many problems in data science require minimizing ranks, and various cost functions have been developed to address this problem[31]. Some of them, in particular the Normalized Discounted Cumulative Gain[32] (NDCG), can readily be employed in multispecies DNNs[33].

Here, we use multispecies DNNs to model the distributions of large numbers of species based on citizen science data. Using occurrence observations of 2477 vascular plant species in Switzerland, we demonstrate that multispecies DNNs can accurately predict observation probabilities of thousands of species, both seasonally resolved and across large spatial scales. To do so, we jointly model all quality-filtered observations of the National Data and Information Center on the Swiss Flora (InfoFlora, www.infoflora.ch, Supplementary Fig. 1). We compile two data sets of different spatial resolution and use them for different parts of the analysis, as indicated in brackets below. The low-resolution data set consists of 4.6 million observations and ten environmental predictors at $100 \times 100$ m resolution while the high-

resolution data set includes 6.7 million observations and 18 environmental predictors at $25 \times 25$ m resolution. At both resolutions we complement environmental predictors with two seasonal predictors based on a sine-cosine mapping of day of year[20], in order to also model observation probability in response to time (daylight hours). We compare the performance of DNNs trained with CEL and the NDCG cost functions to the performance of stacked predictions of SDMs (SSDMs) that were trained with the same environmental but without seasonal predictors (low-resolution data). To highlight the insights that can be derived from multispecies DNNs, we focus on three aspects that are difficult to analyze with traditional approaches: (1) mapping variations in phenology (high-resolution data); (2) mapping potentially dominant tree species (high-resolution data); and (3) projecting spatiotemporal distributions of graminoid species under climate change (low-resolution data).

## Results
### Performance
Compared to SSDMs, DNNs made better predictions to left-out citizen science observations, with similar performance for the DNNs trained with the CEL and the NDCG cost functions, and even slightly higher performance for ensembled DNN predictions. We measured predictive performance based on the ranks of observed species in model predictions for 2465 species for which both DNN and SDM predictions existed. Ranks close to one thereby represent highest performance and ranks close to the maximum of 2465 species represent lowest performance (Fig. 1). We used paired Wilcoxon tests and the Holm correction[34] for multiple comparisons to test for significant differences between models. Note that this paired test evaluates the distribution of score differences per observation, while the boxplots in Fig. 1 display the distribution of scores across observations. According to the Wilcoxon tests, significant differences ($p \le 0.001$, $n = 12,325$) were found between all pairwise comparisons except for the CEL DNN and the NDCG DNN, with lowest ranks for ensembled DNN predictions (combined by square root of the geometric mean), followed by CEL DNN predictions, NDCG DNN predictions, and SSDM predictions. Median ranks, however, were quite similar for DNNs, with 73, 73, and 71, for the CEL DNN, the NDCG DNN, and the DNN ensemble, respectively, and, with 169, distinctly higher for SSDM predictions. The DNN ensemble trained with the best high-resolution data available achieved a median rank of 41. Training the low-resolution CEL DNN without seasonal predictors, i.e., with the same information as SSDMs, still yielded distinctly higher performance, with a median weighted rank of 91 (Supplementary Fig. 2). The results remained qualitatively similar, when performance was measured with top-1, top-5, top-10, and top-20 accuracy (Supplementary Fig. 2).

DNNs also showed a higher capacity than SSDMs to predict individual species' distributions and in particular community composition. As an independent test set, we used 1489 regularly distributed plant community inventories from the Swiss Biodiversity Monitoring program (www.biodiversitymonitoring.ch). These regularly distributed surveys (Supplementary Fig. 1) covered 1345, mainly common species, and thus somewhat more than half the 2465 species as evaluated above. Wilcoxon tests found no significant differences in predicting species distributions for SSDM predictions and annual summaries (see methods) of NDCG DNN predictions, but CEL DNN predictions and in particular DNN ensemble predictions were significantly better (Fig. 1b; $p \le 0.001$; $n = 1345$), with corresponding median species-by-species area under the curve[35] (AUC) of 0.937, 0.937, 0.942, and 0.945. The advantage of DNNs was even higher when predicting community composition. Median site-by-site AUC was 0.964, 0.974, 0.974, and 0.976, for SSDMs, the NDCG DNN, the CEL DNN, and the DNN ensemble, respectively (Fig. 1c), with significant differences ($p \le 0.001$; $n = 1489$) for all comparisons but CEL DNN and NDCG DNN. The DNN ensemble trained with the best high-resolution data available achieved

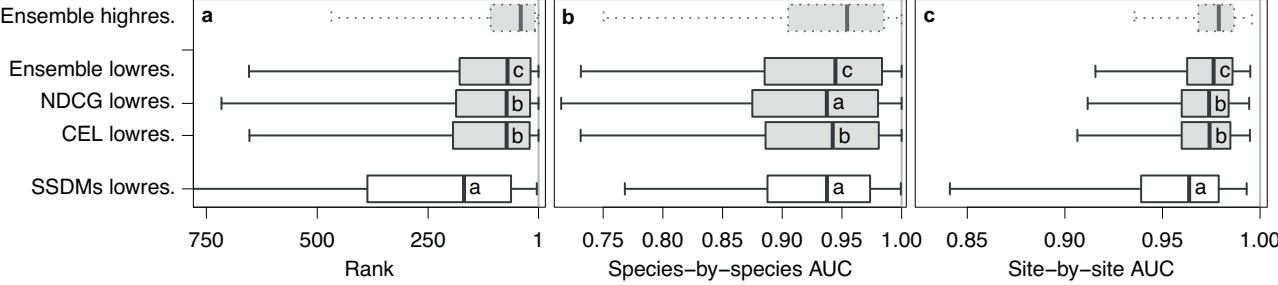

**Fig. 1 | Performance of deep neural networks (DNNs) trained with different cost functions and data sets in comparison to stacked species distribution models (SSDMs). a** Ranks inferred from a left-out test set of $n = 12,325$ citizen science observations. **b** species-by-species AUC measuring the performance of predicting the distributions of 1345 individual species based on independent survey data. **c** site-by-site AUC measuring the performance of predicting the composition of communities at 1489 sites based on independent survey data. Note we weighted the taxonomically balanced citizen science test set (five observations per species) with the number of training observations per species to obtain scores that represent typical field observations (see methods). Boxes of DNNs are shown in gray and boxes of SSDMs are shown in white. Results of the DNN trained with high-resolution data are shown with dashed lines. Central lines in the boxplots indicate medians, boxes indicate interquartile ranges, and whiskers indicate 2.5 and 97.5 percentiles; NDCG refers to Normalized Discounted Cumulative Gain; CEL refers to Cross Entropy Loss; and lowres. and highres. represent low resolution and high resolution, respectively. Letters superimposed on boxes represent results of two-sided paired Wilcoxon tests, with no significant differences for models sharing the same letter and significant differences otherwise. The exact $p$-values can be found in Supplementary Table 3. Source data are provided as a Source Data file.

even higher performance with a species-by-species AUC of 0.954 and a site-by-site AUC of 0.979.

### Predicting spatial variations in species' phenology
Plant phenology is the study of seasonally recurring phenomena in plant development, which are important indicators of ecosystem functions and their susceptibility to climate change[36]. Despite increasing attention, our capacity of modeling phenological events is still limited[36,37]. We explored the spatial patterns in the seasonal variation of observation probability and their links to phenology for a selected set of common species, by mapping the timing of peak observation probability ($t_{p\_max}$) based on predictions of the high-resolution DNN ensemble. To this end, for each pixel we made daily predictions of observation probabilities from March to September and identified on which day observation probability was highest (see methods).

At the scale of Switzerland $t_{p\_max}$ showed distinct spatial patterns for the evaluated species. $t_{p\_max}$ of bugle (*Ajuga reptans*) started in April at lower elevations, while at higher elevations in the Prealps and the Jura mountains it was around early June (Fig. 2a). The case was similar for common twayblade (*Listera ovata*) that showed an even higher plasticity in $t_{p\_max}$ (Fig. 2b). Also, for the common dandelion (*Taraxacum officinale* aggr.) $t_{p\_max}$ was quite variable (Fig. 2c). Yet, in some regions of the Swiss plateau, the pattern was more scattered than it was for the former two species. The seasonal progression of observation probability at a few selected locations indicates that this scattering results from double peaks in observation probability: on the Swiss plateau observation probability peaked twice, once around the end of April and once in August. This phenology was distinct from the phenology in the high-elevation valleys of the eastern alps, where $t_{p\_max}$ was between end of May and June, but also form the phenology in the lowland areas of the central valley and southern Ticino, where $t_{p\_max}$ was in early April without a second peak later in the season.

$t_{p\_max}$ also showed striking local patterns in a mountainous $5 \times 5$ km square north of Grindelwald. For early-purple orchid (*Orchis mascula*, Fig. 2d), false aster (*Aster bellidiastrum*, Fig. 2e), and slender bedstraw (*Galium pumilum* aggr., Fig. 2f) $t_{p\_max}$ delayed rapidly with elevation. To a lesser degree $t_{p\_max}$ also responded to other terrain properties, such as slope and exposition, as in the case of early-purple orchid, or troughs, as in the case of false aster.

Predictions of $t_{p\_max}$ matched well with dates on which the selected species were seen blooming. We used the minor fraction of InfoFlora observations for which phenology information was available

and classified as 'full bloom' to compare their dates to corresponding predictions of $t_{p\_max}$. For the species depicted above, median difference between observation dates and $t_{p\_max}$ ranged from 1 to 8, while Spearman rank correlation ($r$) ranged from 0.39 to 0.76 (Table 1). Across all species with more than 10,000 training observations and 30 phenology observations of the category 'full bloom', scores were more variable (Supplementary Table 1), with low scores, e.g., for invasive neophytes that are heavily reported to InfoFlora. Nevertheless, median $r$ across these common species was 0.30, and median absolute bias was 8 days.

### Predicting potentially dominant species
For many problems in ecology, ecosystem management, and conservation knowledge is necessary not only on species' habitat suitability but also on potential dominance, especially for structurally important species like trees[38]. We estimated potential dominance for 37 tree species form predictions of the high-resolution DNN ensemble by reweighting observation probabilities to represent Braun-Blanquet[39] cover-abundance scores in the canopy. This reweighting enforced species-specific sums of predicted probabilities across 12,911 sites of the Swiss forest vegetation database[40] to be equal to sums of observed Braun-Blanquet scores, which also corrected for taxonomic reporting bias.

Across Switzerland, Norway spruce (*Picea abies*), especially in the Alps, and European beech (*Fagus sylvatica*), especially on the Swiss Plateau, were predicted to potentially dominate most often, in 41.8% and 37.6% of the wooded area, respectively (Fig. 3a). Other species were predicted to potentially dominate in a narrower range of environmental conditions, for example European larch (*Larix decidua*, 4.5%) and Swiss stone pine (*Pinus cembra*, 3.5%) in the alps, Scots pine (*Pinus sylvestris*, 4.3%) in the Central valley and the upper Rhine valley, and sweet chestnut (*Castanea sativa*, 1.7%) on low-elevation slopes in Ticino.

At the local scale, a turnover in potentially dominant tree species was predicted along gradients of elevation and exposition. On the steep, south-facing slope north of Walensee, European beech and Scots pine were predicted to potentially dominate from the lake shore to intermediate elevations (Fig. 3b), while Norway spruce and silver fir (*Abies alba*) were predicted to take over potential dominance at relatively high elevations but reached further downhill on the north-facing side, behind the ridge. On the eastern side of the Riviera valley, potentially dominant tree species showed a distinct horizontal layering (Fig. 3c). On the valley bottom, European ash (*Fraxinus excelsior*) and

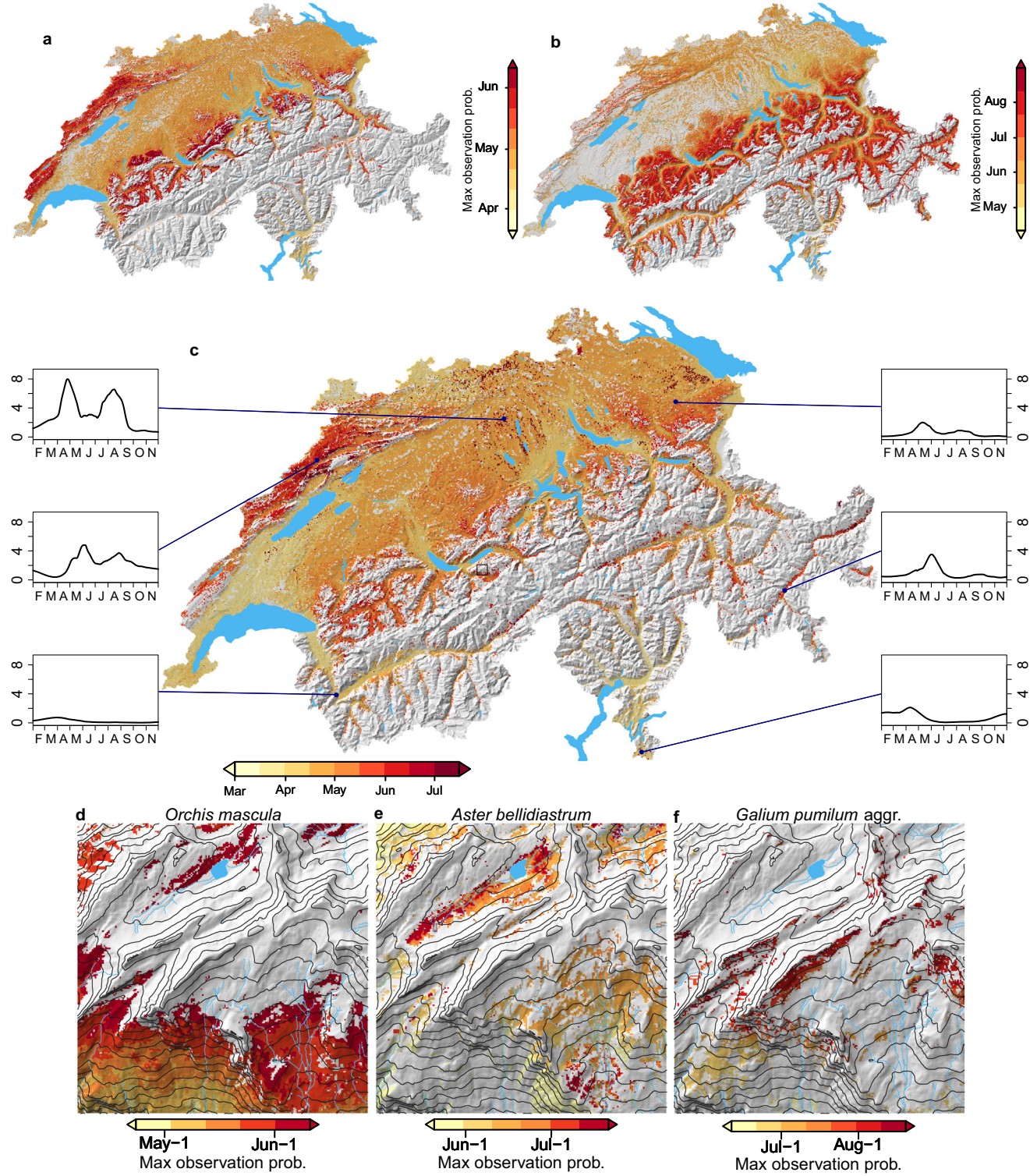

**Fig. 2 | Seasonal maxima of observation probability for selected species.** Panels **a**–**c** show the timing of highest observation probability across the distribution ranges of *Ajuga reptans*, *Listera ovata* and *Taraxacum officinale* aggr., respectively. Shown are pixels with a maximum daily observation probability ≥0.01 between March and September. Insets in panel **c** illustrate the seasonal evolution of observation probability (in percent, from February to November, no smoothing applied) at six selected locations within the distribution range of *T. officinale* aggr. Panels **d**–**f** also show timing of peak observation probability, but for a 5 × 5 km square in the region of Faulhorn (outlined on the map in panel **c**) for *Orchis mascula*, *Aster bellidiastrum*, and *Galium pumilum* aggr., respectively. Maps were created using the open-source R software (see methods).

other species potentially dominated; at the lower slopes it was sweet chestnut intermixed with downy oak (*Quercus pubescens*); higher up European beech took over for an elevation band of several hundred meters before being displaced by Norway spruce and silver fir.

Predictions of potentially dominant species showed a reasonable agreement with observations from the National Forest Inventory[41]. For 1867 regularly-distributed sites, we compared the species with the highest (bias-corrected) observation probability to the species with

**Table 1 | Predictive performance of the timing of blooming for the species shown in Fig. 2**

| Species | n | Spearman r | Bias (days) |
|---|---|---|---|
| *Ajuga reptans* | 144 | 0.62 | 4 |
| *Aster bellidiastrum* | 77 | 0.59 | 4 |
| *Galium pumilum* aggr. | 28 | 0.39 | 1 |
| *Listera ovata* | 3313 | 0.76 | 1 |
| *Orchis mascula* | 4491 | 0.52 | 8 |
| *Taraxacum officinale* aggr. | 161 | 0.41 | 3 |

*n* refers to number of phenology observations for testing; *r* refers to correlation coefficient.
Source data are provided as a Source Data file.

most observed individuals with ≥12 cm stem diameter at breast height. Overall top-1 accuracy of bias-corrected predictions was 0.50. $F_1$ scores were comparably high for the common species Norway spruce and European beech, but also for rarer alpine species like Swiss stone pine (Table 2). On the other hand, bias-corrected predictions were unsuccessful for widespread but rarely dominant lowland species, such as silver fir, sycamore maple (*Acer pseudoplatanus*), European ash, sessile oak (*Quercus petraea*), and silver birch (*Betula pendula* aggr.). When predicting potentially dominant tree species without prior bias correction, overall accuracy was lower (0.38) but $F_1$ scores increased for most of the less common species, especially for silver fir and European larch, whose predictions were down weighted with bias correction.

**Predicting climate-change impact**

Projections of phenology and potential dominance can also enrich assessments of climate change impact, which may allow for better-informed mitigation strategies. Using the low-resolution DNN ensemble, we predicted potential dominance of graminoid species along an elevational gradient (400–2900 m asl.) in the canton of Vaud under current conditions and under conditions expected in 2060, assuming RCP4.5[42]. We reweighted predictions as for the tree species but based on 23,919 community plots from the dry meadows and pastures initiative[43], and we sub-selected the twelve most-frequent potential dominators.

In May and June, under current conditions, the potentially most dominant species were predicted to be bulbous oat grass (*Arrhenatherum elatius*), erect brome (*Bromus erectus*), and blue moor-grass (*Sesleria caerula*) for the valley bottom, the lower slopes and the upper slopes, respectively (Fig. 4a, b). In July and August, erect brome still potentially dominated, while in the valley it was intermediate wheatgrass (*Elymus hispidus*) and on the upper slopes several species were predicted to partially dominate, including evergreen sedge (*Carex sempervirens*) at intermediate elevations, and purple fescue (*Festuca violacea* aggr.) at high elevations (Fig. 4c, d).

For 2060, potential dominance was projected to change, and species to shift their ranges (defined as gravity centers of dominance probability) towards higher elevations and to advance $t_{p\_max}$ (Fig. 4e–h). Intermediate wheatgrass was projected to take over potential dominance already in May at lowest elevations. Note that at these elevations warming led to predictions beyond training range (gray-shaded area in Fig. 4e–h), which need careful interpretation. At higher elevations the potential dominance of purple fescue disappeared, and evergreen sedge was almost entirely displaced by red fescue (*Festuca rubra* aggr.) and crested dog's-tail (*Cynosurus cristatus*). Among the species projected to move upward was blue moor-grass that was predicted to potentially dominate with elevated probability at highest elevations already in May.

**Discussion**

Our results contribute to a rapidly growing body of evidence highlighting that DNNs can efficiently learn the distributions of thousands of species based on millions of citizen science observations, without the need of sacrificing observations to spatial bias correction. In agreement with recent comparisons on plants[23,26] and other taxa[44,45], we found multispecies DNNs to perform better than SSDMs in all comparisons. The advantage of DNNs, thereby, was especially pronounced in ranking observed species and in predicting community composition, which are both multispecies problems. Performance differences between the CEL and the NDCG DNN were minor, but there was a consistent performance gain when ensembling them. The alternative design of the NDCG cost function, with its less strict rejection of unobserved classes, did not yield model fits that were superior compared to those trained with the CEL, but it led fits that behaved differently. Ensembling the predictions of models trained with both cost functions appears to efficiently combine their advantages, outperforming CEL DNNs in a similar way as the approaches reported in ref. 29. More importantly, however, multispecies DNNs offer new and promising perspectives to ecological research.

Firstly, they can resolve spatial variations in phenology. For common forb species we have identified the timing of peak observation probability within pixels and revealed striking spatial patterns in detail. Our results suggested interesting phenomena, such as double peaks in the blooming of *T. officinale*, which have also been observed in other regions[46]. The correspondence of predicted $t_{p\_max}$ with observed phenology was satisfying, given the extended duration of flowering periods, their substantial interannual variations[47], and the indirect nature of our approach. However, our estimates of $t_{p\_max}$ are only representative for Switzerland and may not cover the full plasticity of the species, much like empirical estimates of ecological niches may be truncated if the species' distribution extends beyond the study area[48]. Future efforts may further improve estimates of $t_{p\_max}$ by filtering training observations for blooming individuals as identified from auxiliary images[49] and by using annual climate data. Understanding the spatial variations in species' phenology can improve, for example, our understanding of speciation processes through reproductive isolation[50], and the impact of phenological mismatches on the spatial variation of plant-pollinator networks[51].

Secondly, multispecies DNNs can jointly predict observation probabilities for many species. We have generated a nationwide map of potentially dominant canopy-forming trees that is more detailed than existing alternatives[52,53] and that corresponds reasonably well to observations, given the relatively crude input on vegetation structure and that 45% of Swiss forests are not in a natural state[54]. Such a map cannot be created by stacking predictions from individual species' models (Supplementary Fig. 3), and it would likely be less detailed without ubiquitous citizen science observations. Still, for some widespread species that rarely dominate large stands, such as *Acer pseudoplatanus*, *Betula pendula*, *Fraxinus excelsior*, and *Quercus petraea*, predictions were rather poor. In addition to environmental conditions, the distribution of these species is determined by the successional stage of the forest[55], by management intervention, or by demographic processes, which may not have been captured in sufficient detail by our set of environmental predictors. However, for many of the less-common species, $F_1$ scores could be improved somewhat when not correcting for taxonomic reporting bias. While these corrections improved overall accuracy from 0.38 to 0.50, primarily because of an increased weight for observation probabilities of Norway spruce, they reduced $F_1$ scores of many of the less-common species. When mapping such species individually, the underproportioned attention Norway spruce and European beech receive from citizen scientists thus appears to be beneficial. Predictive performance of less common species may further improve if DNNs are trained exclusively for canopy forming tree species, if a temporal match between observations and predictors was enforced, and in particular if more detailed remote sensing predictors were used[21,38]. Maps of potentially dominant species may be useful for forest rangers to identify locally competitive species[56], but comparable observation probabilities are also

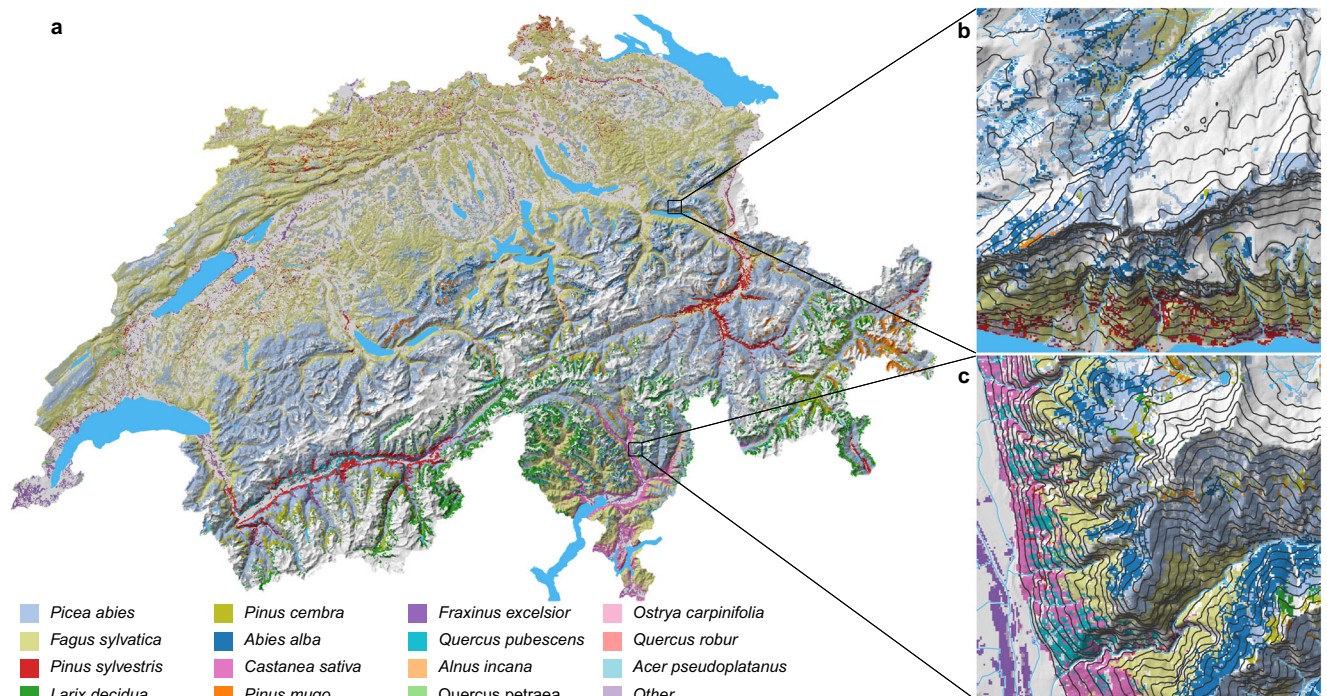

**Fig. 3 | Potentially dominant canopy-forming tree species in wooded areas of Switzerland. a** For the entire country; **b** for a selected 5 × 5 km square near Quinten; **c** for a selected 5 × 5 km square near Biasca. Colors represent species with the highest weighted observation probability averaged from February to November (as indicated in the legend at the figure bottom). We compared observation probabilities of 37 tree species, as distinguished by the Swiss forest vegetation database[40], and masked pixels with land cover classes without trees (see methods). Light blue gray represents *Picea abies*; chalky represents *Fagus sylvatica*; red represents *Pinus sylvestris*; lime green represents *Larix decidua*; pea green represents *Pinus cembra*; dark blue represents *Abies alba*; orcid pink represents *Castanea sativa*; dark orange represents *Pinus mugo*; amethyst purple represents *Fraxinus excelsior*; topaz blue represents *Quercus pubescens*; light orange represents *Alnus incana*; light green represents *Quercus petraea*; powder pink represents *Ostrya carpinifolia*; light salmon pink represents *Quercus robur*; morning glory blue represents *Acer pseudoplatanus*; and wisteria purple represents other species. Maps were created using the open-source R software (see methods).

advantageous in other contexts. They can, for example, be used to quality control citizen science observations or to improve image-based plant species identification[19,20].

The third perspective emerges from projecting climate-change impact on phenology and potential dominance. In addition to moving towards higher elevations, our DNN projected, e.g., intermediate wheatgrass to advance $t_{p\_max}$ in the season, which matches well with the general expectation of advancing phenology in European grasslands[57]. Modeling climate change impact on phenology at the species level allows quantifying to which extent plants can cope by shifting their growing season, which may improve their resilience[58], and it allows assessing where phenological mismatches of interacting species may arise, for example between plants and pollinators[59]. Moreover, when potential dominance can be estimated, multispecies DNNs allow projecting how it may change, as we have shown for evergreen sedge and red fescue at intermediate-to-high elevations. And when such turnovers affect structurally important species, consequences at the ecosystem level may be dire. While our model set-up permits detailed insight into these important aspects of distributional change, we would like to emphasize that it was not optimized for climate change impact assessments, which need careful design to limit uncertainties from model extrapolation[60]. Firstly, our predictors on vegetation height and greenness allowed accurate predictions under current conditions, but for future projections it would be safer to remove them and avoid the assumption that vegetation does not change in the coming decades, which we made in our simple illustration. Secondly, for climate change impact projections it would be preferable to consider observations from species' entire ranges rather than from Switzerland only, to reduce problems from extrapolation and niche truncation as much as possible[61,62]. Finally, as many machine-learning techniques, standard DNNs are not designed for extrapolation, and it may be more meaningful to rely on ongoing developments with better informed extrapolation behavior[63]. Nevertheless, accounting for phenology and potential dominance may critically refine climate-change impact assessments that so far have often focused on range shifts[64].

Every year, citizen scientists report millions of species observations which are urgently needed to cope with the ongoing biodiversity crisis[5] but are also biased and challenging to analyze. Multispecies DNNs are not only capable of skillfully modeling such data, but they also offer unprecedented insight into the spatial patterns of species' phenology and potential dominance, which allows comprehensive prognoses of climate-change impact. The approach is applicable broadly beyond the examples shown here, for example to quantify spatiotemporal variations of biodiversity patterns, or to investigate climate change-related habitat turnover. Sufficiently high observation numbers are a mandatory prerequisite to successfully build taxon-wide multispecies DNNs, which is why they may currently be most advantageous in well-sampled regions. But given that citizen science contributions to biodiversity monitoring keep growing[2], and that we have only started exploring this method, chances are high that it will become a standard item in the toolbox of tomorrows macroecologists.

## Methods
### Overview
We trained different versions of multispecies DNNs, assessed their performance, and explored their seasonal and spatial predictions under current and future climate conditions. Different versions of low-resolution (100 × 100 m) multispecies DNNs were trained, using two cost functions and several predictor sets, and their performance was compared to stacked species distribution models that were fitted at the same low resolution. Additionally, we trained and evaluated high-

**Table 2 | Predictive performance for the 13 most-commonly dominating canopy tree species for predictions corrected for taxonomic reporting bias and for uncorrected (raw) predictions**

| Species | n | Correction weight | Bias corrected | | Raw | |
|---|---|---|---|---|---|---|
| | | | F₁ | Correct (#) | F₁ | Correct (#) |
| *Picea abies* | 732 | 1.86 | 0.67 | 496 | 0.45 | 238 |
| *Fagus sylvatica* | 407 | 2.11 | 0.56 | 354 | 0.57 | 260 |
| *Abies alba* | 207 | 0.79 | 0.03 | 4 | 0.32 | 107 |
| *Larix decidua* | 114 | 0.39 | 0.25 | 17 | 0.46 | 49 |
| *Acer pseudoplatanus* | 80 | 1.07 | 0 | 0 | 0.02 | 1 |
| *Fraxinus excelsior* | 68 | 1.28 | 0.07 | 3 | 0.09 | 4 |
| *Castanea sativa* | 40 | 1.29 | 0.5 | 17 | 0.58 | 21 |
| *Pinus sylvestris* | 32 | 2.23 | 0.24 | 14 | 0.17 | 5 |
| *Betula pendula* aggr. | 25 | 0.2 | 0 | 0 | 0.06 | 1 |
| *Alnus incana* | 21 | 1.8 | 0.24 | 4 | 0.16 | 2 |
| *Pinus mugo* | 20 | 0.77 | 0.15 | 2 | 0.3 | 5 |
| *Quercus petraea* | 19 | 1.92 | 0 | 0 | 0 | 0 |
| *Pinus cembra* | 17 | 0.43 | 0.44 | 7 | 0.45 | 13 |

*n* refers to number of test observations; F₁ score is the harmonic mean of precision and recall; and correct (#) refers to number of correctly classified observations per species. Source data are provided as a Source Data file.

resolution (25 × 25 m) DNNs and used them to map the timing of seasonal maxima in observation probability ($t_{p\_max}$), as well as potentially dominant trees at different spatial scales. Finally, using low-resolution DNNs, we investigated how predicted observation probabilities may be altered by 2060, assuming an intermediate emission scenario.

## Data

**Observations.** We worked with five sets of observational data: a main set of opportunistic citizen science observations for model training, and four sets of expert-based survey data for independent validation and correction of taxonomic reporting bias. The opportunistic data set was provided by the National Data and Information Center on the Swiss Flora (InfoFlora, www.infoflora.ch) and contained plant observations in Switzerland that were primarily made by citizen scientists. Observations were made during recent decades, with continuously increasing numbers per year and 80% of contributions after 1998 (Supplementary Fig. 4). We filtered these data for observations made in 1971 or more recently, for taxon labels at the level of species aggregates or finer, for identifications with high confidence as indicated by the users, for coordinates within Switzerland that are plausible for the species (as assessed by InfoFlora), and for a coordinate uncertainty of no more than 100 m. Then, we removed duplicates and pooled observations with subspecies-level labels with those of the corresponding species, and observations with species-level labels with those of the corresponding species aggregates (for the 12% of species that are part of species aggregates). For simplicity, we jointly referred to species and species aggregates as 'species' throughout the main text. We used two versions of InfoFlora data: for low-resolution DNNs and for stacked SDMs we worked with an extraction that was made in April 2019, consisting of 5.1 million records after filtering. For multi-species DNNs we further filtered this data set for the availability of an observation date. In this step, we also excluded observations with time stamps representing the last or the first second of the year, which originated from observation time information with annual resolution. Filtering for observation date reduced the data set to 4.6 M observations. For the high-resolution analysis, we worked with a more recent

extraction (November 2022) and applied a stricter 25-m-threshold to coordinate uncertainty and the same filter for observation date, resulting in 6.7 M filtered records.

We partitioned the InfoFlora observations into training and test data (Supplementary Fig. 1) and filtered for taxa with a minimum number of occurrences. Test data consisted of five randomly sampled observations per taxon with observation date and no more than 25 m coordinate uncertainty. We sampled these test observations from the more recent version of the data set, wherever possible with observation dates after April 2019 (99.8% of cases) and where not possible with older observation dates. With this selection approach we ensured a fair comparison between low-resolution models, which was the main aim of this study, but we also created small temporal mismatch between training and test data for low-resolution models, which may be a disadvantage compared to the set-up for high-resolution models. Training data of both versions of the opportunistic data set were then defined as all observations that were not part of the test set, except in case of stacked species distribution models, where we used all observations of the older version of the data set as training data (meaning that SDMs have seen 0.02% of test observations during training). In addition to five test observations, we expected at least 20 training observations in both versions of the data set for a taxon to be considered, which was the case for 2477 species and species aggregates.

For the low-resolution DNNs, we additionally created a validation data set that consisted of five observations per taxon, as did the test set. These validation observations were drawn randomly from the full low-resolution data, independent of the partitioning in training and test set. Validation runs were conducted prior to the training of the main DNNs, where observations from the validation set may either be in the training or the test set. InfoFlora observations were accessed via application programming interface, using the R environment (version 4.0.4)[65] and the R package rjson (version 0.2.20)[66].

Surveys from the Swiss forest vegetation database and the dry meadows and pastures initiative were used to correct for taxonomic reporting bias. The Swiss forest vegetation database[40] reported the cover-abundance of 41 tree and shrub species at three vertical layers for 14,860 survey sites within Switzerland and in neighboring regions. We filtered the data for surveys within Switzerland with full coverage of environmental data (see below) and summed species-level information to species aggregates, where applicable. Semi-quantitative cover-abundance information from the canopy-layer was translated from the original Braun-Blanquet scheme[39] to percentages, following ref. 67. After filtering and aggregation, cover-abundance information was available for 37 taxa at 12,911 survey sites (Supplementary Fig. 1). The dry meadows and pastures initiative[43] was run by the Swiss Federal Office for Environment and consists of cover-abundance information for almost 24,000 surveyed grassland communities. We focused on 91 graminoid taxa that are dominant and influence the appearance of at least one grassland, wetland or cropland habitat type in Switzerland, according to the TypoCH vegetation classification[68], and conducted the same aggregation and filtering steps as for the surveys from the Swiss forest vegetation database. After processing, cover-abundance information was available for 23,919 surveys (Supplementary Fig. 1).

Expert-based surveys from the Swiss Biodiversity Monitoring Program[69] (BDM, www.biodiversitymonitoring.ch), and from the National Forest Inventory[41] (NFI, www.lfi.ch) were used for independent validation. Survey data from the BDM program were used to validate predicted distributions of individual species and community composition. We considered the BDM indicator Z9 (species diversity in habitats), containing 1562 sites that are regularly distributed across Switzerland. Between 2000 and 2021 each of these 10 m² areas was revisited three to four times and all plant taxa present were recorded. Taxa were mostly identified at the species level, but identifications at the level of species aggregates were also common. We considered a taxon present at a site if it was observed in at least one revisitation.

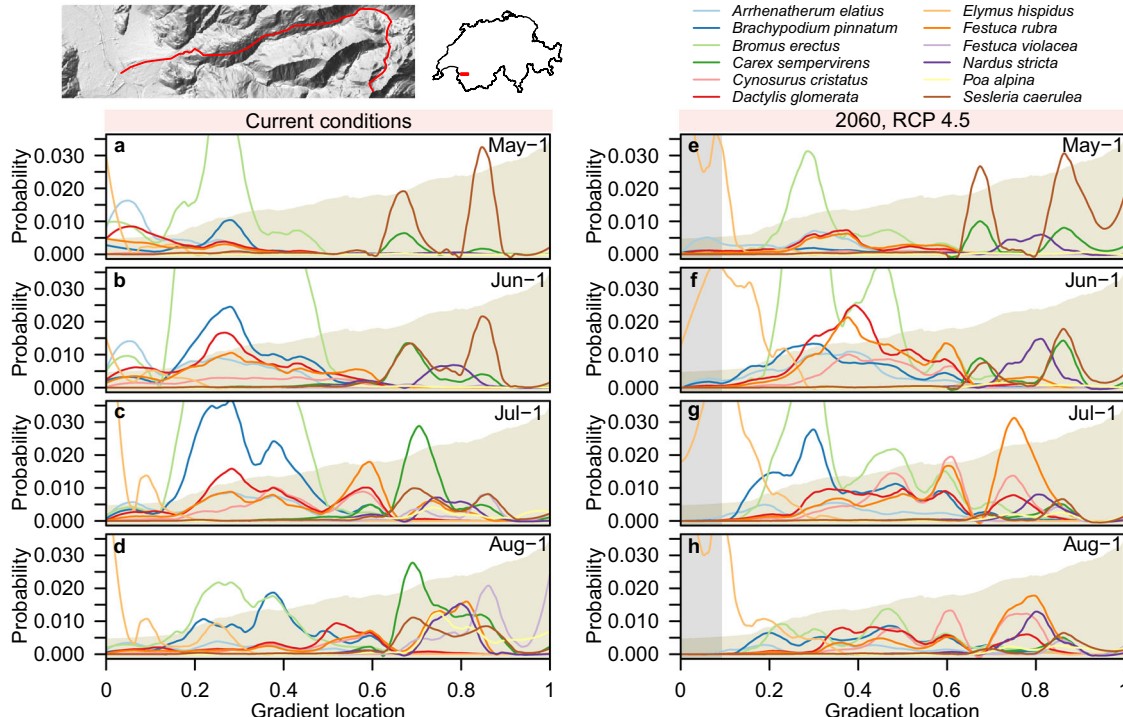

**Fig. 4 | Reweighted observation probabilities of graminoid taxa along an elevational gradient in the canton of Vaud (see inset maps).** Panels **a–d** illustrate bias-corrected observation probabilities along the gradient for the twelve most likely observed graminoid species (see legend) under current conditions for May-1, June-1, July-1, and August-1, respectively. Panels **e–h** illustrate equivalent observation probabilities for 2060, assuming the emission scenario RCP4.5. Probabilities along the gradient are represented as local polynomial regression fits (see methods). Yellowish shading in the background represents the elevation profile ranging from 400 to 2900 m asl. Gray-shaded area in panels **e–h** indicates extrapolations beyond training range. For an SSDM-based version see Supplementary Fig. 7. Light blue represents *Arrhenatherum elatius*; dark blue represents *Brachypodium pinnatum*; light green represents *Bromus erectus*; dark green represents *Carex sempervirens*; light red represents *Cynosurus cristatus*; dark red represents *Dactylis glomerata*; light orange represents *Elymus hispidus*; dark orange represents *Festuca rubra*; light purple represents *Festuca violacea*; dark purple represents *Nardus stricta*; yellow represents *Poa alpina*; and brown represents *Sesleria caerulea*. Source data are provided as a Source Data file. Maps were created using the open-source R software, with information on the country borders originating from the SwissTLM3D topographic landscape model (see methods).

Where applicable, we pooled species to aggregates, yielding 1489 survey sites (Supplementary Fig. 1) with at least one species or species aggregate observed and full environmental data coverage (see below), and 1345 taxa observed across the survey sites. Survey data from the NFI were used to validate predicted potentially dominant tree species. The NFI samples forests on a regular grid with 1.4 km spacing (where forests occur, see Supplementary Fig. 1). The sampling procedure includes the species-level identification of all trees with ≥12 cm stem diameter at breast height in an inner circle covering 200 m², and the identification of all trees with ≥36 cm diameter at breast height within an outer circle covering 500 m². We considered survivor trees from the third (2004–2006) to the fourth (2009–2017) assessment[41] and defined the species with most individuals with ≥12 cm stem diameter at breast height to be dominant. After filtering and pooling for the same 37 species and species aggregates as distinguished by the Swiss forest vegetation database, information on dominant tree taxa at 1867 sites was available.

**Environmental predictors.** We constructed a low-resolution and a high-resolution set of environmental predictors, representing vegetation structure, climate, soil conditions, topography, and season. Two separate sets were necessary to allow for a fair comparison between multispecies DNNs and stacked SDMs (which were trained in 2019), and to include the best data available for the high-resolution analysis. The low-resolution (100 × 100 m) set included 24 environmental variables, projected in the CH1903/LV03 projection system, and two predictors representing season. The low-resolution environmental layers originated from two recent studies investigating the distribution of

Swiss plant species[16,70]. The predictor group representing vegetation structure included canopy height, inner forest density, and mean and standard deviation of the normalized difference vegetation index (NDVI). Canopy height and inner forest density were estimated by the 25th and 95th height percentiles of normalized LiDAR returns above 40 cm[16,71]. Mean and standard deviation of NDVI were calculated from Landsat measurements (https://espa.cr.usgs.gov/) taken during summer months (July to mid-September) between 2007 and 2015. Climate variables describe temperature, precipitation, and solar radiation (direct and diffuse) and were downscaled from reanalysis data[72] to the EarthEnv-DEM90 digital elevation model[73], using the CHELSA approach[74]. For temperature and precipitation, we considered averages and sums, respectively, for June to August, December to February, and for the whole year. For solar radiation, we only considered annual averages. In addition, we included CHELSA-based estimates of frost change frequency and growing degree days (with 5 °C baseline temperature) and plant indicator value-based estimates of continentality and light availability[70] as climatic predictors. Soil conditions are represented by plant indicator value-based estimates of pH, moisture, moisture variability, aeration, humus, and nutrients[70]. Topography variables included terrain wetness index, terrain ruggedness index, and terrain position index, and were derived from the EarthEnv-DEM90 digital elevation model[73]. More details on the generation of these predictor variables are provided in Refs. 16,70. Finally, we used sine-cosine mapping to encode the season of observation (day of year) into two circular variables[20], one of them roughly indicating day length and the other change in day length per day. Collinearity in this set of predictors overall was rather low, except for some predictors related

to temperature and, to a lesser degree, precipitation (Supplementary Fig. 5). In the subsequent analyses predictors were subsampled in order to avoid pairs with high correlations (see below).

The high-resolution (25 × 25 m) predictor set roughly covered the same predictor groups but included some additional variables. High-resolution variables were compiled in the CH1903/LV95 projection system. For vegetation structure, we aggregated the country-wide vegetation height model with 1 m original horizontal resolution[75] to estimate maximum and 25th percentile of vegetation height in each 25 × 25 m cell. In addition, we calculated annual medians and seasonal differences in enhanced vegetation index (EVI) for the period 2018–2021. EVI data was calculated based on 10 × 10 m Sentinel 2 measurements[76] and compiled to analysis-ready annual and seasonal medians by the Swiss Data Cube[77–81]. We removed values outside the valid range (−1 to 1), and reprojected and aggregated the layers to 25 × 25 m by average. Then, we aggregated annual, summer, and winter medians inter-annually by median and calculated the difference between summer and winter median. For a handful of pixels, no valid EVI measurements were available, so we approximated them with bilinear interpolation. We also considered data on canopy mixture (deciduous versus evergreen) in forested areas at 25 × 25 m original resolution[53]. We encoded this information as two variables, one representing deciduous forest cover and one representing evergreen forest cover, setting the values for unforested areas to zero. Related to vegetation structure, we also included information on land cover at 25 × 25 m original resolution[82], encoding the 62 classes distinguished individually as binary factors (one hot encoding). High-resolution temperature and precipitation data were derived from the CHclim25 data set[83–85]. These data were calculated with topography-dependent interpolation of station measurements to 25 m resolution. From the monthly climatologies for the period 1981–2010 we derived annual means and sums, for temperature and precipitation respectively, as well as annual ranges. Moreover, we calculated precipitation sums for the summer months (June to August). Terrain variables were derived by aggregating the SwissAlti3D model, distributed by the Swiss Federal Office of Topography, from the original 2 m resolution to 25 m by average. From the elevation data (not used as predictor) we calculated aspect as well as the SAGA wetness index (using SAGA GIS[86]). In addition, we calculated the terrain ruggedness index for each 25 × 25 m cell (maximum minus minimum elevation across the covered 2 × 2 m cells). To represent solar radiation, soil conditions, and season, we used the same variables as in the low-resolution set, downscaling them, where applicable, with bilinear interpolation. Collinearity in the high-resolution predictor set was low (Supplementary Fig. 6), except for the association of mean annual temperature with annual temperature range ($r = 0.74$) and the association between annual precipitation and summer precipitation ($r = 0.89$).

For mapping, we overlaid information on terrain, lakes, and rivers based on open data sets from Swiss Federal Office of Topography (swisstopo). High-resolution terrain data originated SwissAlti3D digital elevation model. For mapping, we aggregated these data by mean to the appropriate resolution and derived slope and aspect, which, in turn, were used to calculate hill shade. Moreover, for the 5 × 5 km maps, we overlaid contour lines of elevation. Information on the location of lakes and rivers was obtained from the SwissTLM3D model provided by swisstopo. Unless otherwise noted, data were prepared in the R environment (versions 3.6.3 and 4.0.4)[65], using the package terra (versions 1.5-21 and 1.7-71)[87].

### Model configurations
#### Multispecies deep neural networks
**Network architecture and training settings.** We trained multispecies deep neural networks at low (100 × 100 m) and high (25 × 25 m) resolution. DNNs were set up with an input layer of one node per

environmental feature, four residual blocks with twice a linear layer of 380 nodes followed by a rectified linear unit and a dropout layer in between, and an output layer with one node for each of the 2477 species and species aggregates. For low-resolution DNNs, we sub-selected ten environmental predictors from the low-resolution set, as this was also the number used by SDMs (if sufficient presence observations existed for the modeled species, see below) and to limit model complexity when projecting to future conditions. The primary criteria thereby were variable importance in species distribution models, and absolute Pearson correlation coefficients of no more than 0.7 (Supplementary Fig. 5 and Supplementary Table 2). Yet, we made one exception by choosing summer temperature over solar radiation, despite somewhat lower variable importance, to be able to better account for climate change effects. For the high-resolution DNN, we included the whole set of environmental predictors, as (1) only two predictor pairs with absolute Pearson correlation coefficients >0.7 existed (Supplementary Fig. 6), and (2) we did not use the high-resolution DNN for future projections. All predictor variables were rescaled to range from −1 to 1 prior to analysis. For low-resolution DNNs, we conducted a sensitivity analysis with an alternative predictor set consisting of environmental predictors only (no seasonal predictors). We trained all models for a first phase of 50 epochs with equal sampling probabilities of training observations for each taxon (by upsampling rare species), and then for second phase with unmodified sampling probabilities. The number of epochs considered for this second phase was inferred from a preliminary validation run using the low-resolution models (see Supplementary Note 1). For high-resolution DNNs, we chose an initial learning rate of $5 \times 10^{-3}$ for training in both phases, while for low-resolution DNNs it was $10^{-2}$ and $5 \times 10^{-3}$, for the first and the second training phase, respectively. In all cases training was optimized with the stochastic gradient descent method[88] and learning rates were reduced when the loss was plateauing, setting the momentum to 0.9. The batch size was set to 250. DNNs were fitted in the python environment[89] (version 3.8.5) using the PyTorch[90] library (version 1.7.1), the pytorchltr library (version 0.2.1)[91], the pandas library (version 1.1.3)[92], and the numpy library (version 1.19.2)[93]. Training times for low-resolution models were about 76 h and 13 h for the NDCG and the CEL cost functions, respectively, when using a Nvidia RTX 3090 GPU.

**Cost functions.** We used two cost functions to train DNNs. As a baseline, we used the cross-entropy loss. The cross-entropy loss[27] (CEL) function is a standard cost function for multiclassification with deep neural networks. The CEL is computationally efficient and has the attractive property of being a strictly proper scoring function that has a unique minimum when the model predicts true conditional probabilities (which are species-specific observation probabilities under the condition that a species presence has been reported at a certain spatiotemporal location)[28]. However, the CEL is designed for classification problems with mutually exclusive classes, such as vegetation types, and is minimized, at the level of an individual observation, when the full probability mass is allocated to the observed class. When confronted with opportunistic citizen science observations, where multiple species are expected to present but we only have information about the presence of one of them (and no information about the presence or absence of all other species), the CEL implicitly assumes that all but the observed species are absent[29]. This assumption may be correct for most of the species from the Swiss flora at each spatiotemporal location, but it is not for the species that are present but not reported. While employing the CEL to this type of data may still lead to fits that perform reasonably well, cost functions that less strictly reject unobserved classes have been shown to be advantageous[29].

The second cost function we used, the normalized discounted cumulative gain (NDCG), makes no assumption about unobserved

classes. Instead, it recasts the learning problem as finding the most relevant identity of a species that has been observed at a given spatiotemporal location, a common task in many data science applications[31], although less common in ecological modeling. Specifically, we employed the LambdaNDCGLoss1 function of the pytorchltr module[91], version 0.2.1, which is defined as

$$l(\mathbf{s}, \mathbf{y}) = -\sum_{i=1}^{n}\sum_{j=1}^{n} \log_2 c_{\pi_i, \pi_j}, \quad (1)$$

where $n$ is the highest rank evaluated and

$$c_{\pi_i, \pi_j} = \left(\frac{1}{1 + e^{-(s_{\pi_i} - s_{\pi_j})}}\right)^{\frac{G_{\pi_i}}{D_i}}, \quad (2)$$

with $\pi_i$ and $\pi_j$ being indices of the items at rank $i$ and $j$, respectively. $\mathbf{s}$ is a vector of scores returned by the model and $\mathbf{y}$ is a vector of relevance labels. Finally,

$$G_{\pi_i} = \frac{2^{y_{\pi_i}} - 1}{\text{maxDCG}}, \quad (3)$$

with max DCG being the maximum discounted cumulative gain, and $D_i = \log_2(1 + i)$. We set $y_{\pi_o}$, i.e., the relevance label at the index of the observed species ($\pi_o$), to one, and all other values of $\mathbf{y}$ zero. Consequently, for our problem, Eq. (1) simplifies to

$$l(\mathbf{s}, \mathbf{y}) = -\sum_{j=1}^{n} \log_2 c_{\pi_o, \pi_j}. \quad (4)$$

We set the highest rank up to which model predictions are evaluated ($n$) to 500.

**Species distribution models.** Single-species SDMs[11] were built by relating occurrences and pseudoabsences to a subset of predictors selected from the low-resolution set. Three replicates of pseudoabsences (10,000) were generated with the geographic-specific approach and a background of environmentally stratified pseudoabsences[16]. Species observations were thinned so that no more than one observation per $100 \times 100$ m grid cell was considered. Each species was modeled with an ensemble of different statistical techniques[94], using five model algorithms with specific parameterizations: generalized linear models[95] (GLMs) were implemented with second-order polynomials and logit link function; generalized additive models[96] were implemented with thin-plate regression splines with maximum of four degrees of freedom, and logit link function; gradient boosting machines[97,98] were implemented with 5000 trees, an interaction depth of two, a learning rate of 0.001, and a Bernoulli distribution; random forests[99] were implemented with 5000 trees and a minimum node size of one; and maximum entropy models[100] were implemented with all feature types. For all algorithms, the total weights of occurrences and pseudoabsences were set to be equal in model calibrations and models were replicated three times with the different sets of pseudoabsences[101–103]. Ensemble predictions were derived by averaging the probabilistic predictions of all combinations of the five algorithms times three replicates. Species distribution modeling was conducted in the R environment (version 3.6.3), using the packages mgcv (version 1.8-32)[104], randomForest (version 4.6-14)[105], gbm (version 2.1.8)[106], and dismo (version 1.3-2)[107].

For each taxon, we used a variable-selection procedure to choose the best predictors. We tested all feasible combinations of predictors by running GLMs with second-degree polynomials and equal sums of weights for occurrences and pseudoabsences, and ranked them based on adjusted explained deviance[11]. Feasible combinations of predictors thereby were defined based on two criteria. Firstly, all absolute

pairwise Pearson correlations within a predictor set had to be lower than 0.7, so that collinearity was contained within the tolerable range[64,108]. Secondly, the number of predictors was fixed for each taxon, and varied between three and ten as a function of the number of occurrences in the calibration dataset (26–35: three predictors; 36–45: four predictors; 46–55: five predictors; 56–65: six predictors; 66–75: seven predictors, 76–85: eight predictors, 86–95: nine predictors; and >95: ten predictors). This second criterion was implemented to achieve an approximate presence-to-predictor ratio $\geq 10^{109,110}$. Taxa with 8–25 occurrences were modeled using ensembles of small models[111], using the three best predictors selected from the variable selection procedure. Taxa with less than eight occurrences were not modeled. For species distribution modeling, we used the low-resolution training data set as described above, but given the spatial thinning applied, data was only sufficient for consider 2465 of the 2477 species modeled with DNNs.

### Predictive analysis
**Assessment of overall performance.** We validated the predictions of multispecies DNNs and stacked SDMs (SSDMs) on citizen science observations and survey data. All trained model configurations were evaluated against the held-out test set of the InfoFlora observations, comparing ranks of observed species in the predictions, and top-1, top-5, top-10, and top-20 accuracies. As we were interested in the performance of typical field observations and our test set was taxonomically stratified, we calculated weighted means of these statistics, using taxon-wise numbers of training observations as weights. We validated both the weighted (after the first 50 training epochs with taxonomically stratified sampling) and the unweighted (after full training) versions of multispecies DNNs. We compared the predictive performance for 2465 taxa for which multispecies DNN and SSDM predictions existed with five observations per taxon, resulting in 12,325 observations. We tested for significant differences between models using two-sided, paired Wilcoxon tests and the Holm correction[34] for multiple comparisons as implemented in the R package rstatix (version 0.7.2)[112].

Multispecies DNNs with the full set of predictors and SSDMs were also evaluated against data from the BDM survey which were not used for training. To this end, we first generated probabilistic predictions for each cell in the taxon-by-site presence/absence matrix. To do so for multispecies DNNs, we first computed observation probability (i.e., applying the softmax function to the raw model predictions) for each day from April to September. Then, we summarized these predictions by their 90th percentile. We used the 90th percentile rather than the maximum, in order to have a robust estimator. Next, for each taxon individually (i.e., for each column), we calculated the area under the receiver operating curve[35] (AUC), indicating the capacity of the models to discriminate between locations where species are present and locations where species are absent. This assessment was based on 1345 taxa which were present in the test data. Similarly, we calculated AUC for each site individually (i.e., for each row), indicating the capacity of the models to discriminate species present at a site from species absent at a site. Site-by-site AUC was calculated based on the predictions for all 2465 taxa for which multispecies DNN and SSDM predictions existed, including those that were never observed in the BDM survey.

**Timing of seasonal maxima in observation probability.** We summarized seasonal observation probabilities pixel-wise by the timing of their maxima ($t_{p\_max}$). For each pixel, we first predicted observation probabilities of the selected species with the high-resolution DNN for each day from March to September. Then, we smoothed the seasonal response curves with a running mean, using the AvgPool1d function of the torch module with a kernel size of 22 days. The curves were smoothed with running means of roughly three weeks to remove

possible short-term fluctuations while still being flexible enough to capture seasonal peaks. Short-term fluctuations existed in rare cases in uncommon habitats, such as steep, rocky slopes, especially when observation probabilities for the focal species were comparably low. Next, we identified the modes of the smoothed response curves, and extracted the timing of the highest mode. Finally, we masked pixels with smoothed maxima below 0.01 and those for which no modes could be identified, as these pixels were assumed to be unsuitable for the species. For the nationwide maps, timings of peak observation probability were aggregated to 200 m spatial resolution by median.

We evaluated how well $t_{p\_max}$ corresponded to the date of Info-Flora observations for which observed individuals were reported to be in full bloom. For a fraction of InfoFlora observations information on phenology state existed, most of which reporting the category 'full bloom' for the observed individuals. We evaluated phenology for the common forbs *Ajuga reptans*, *Aster bellidiastrum*, *Galium pumilum* aggr., *Listera ovata*, *Orchis mascula*, and *Taraxacum officinale* aggr. (shown in Fig. 2), as well as for all other species with at least 10,000 training observations and 30 observations with phenology information (Supplementary Table 1). To this end, we filtered observations with phenology state based on the same criteria as high-resolution training observations and matched them with predictions of $t_{p\_max}$ as described above. Then, we calculated the median difference between $t_{p\_max}$ and observation date (bias), and Spearman rank correlation to measure how well the predicted spatial patterns of $t_{p\_max}$ correspond to observed patterns. Note that although we considered observations for validation that were also used for training, we evaluated the predictions for an unseen attribute (phenological state). Moreover, with 0.5% median, species-specific fraction of training observations used for phenology validation was minor and thus had a negligible effect on the model fit (see Supplementary Table 1).

**Mapping potentially dominant tree species.** In order to map potentially dominant tree species, we first identified pixels with land cover classes including trees and shrubs, i.e., 'normal dense forests', 'forest stripes, edges', 'forest fresh cuts', 'devastated forests', 'open forests (on agricultural areas)', 'open forest (on unproductive areas)', 'brush forest', 'groves, hedges', 'clusters of trees (on agricultural areas)', 'trees in unproductive areas', 'scrub vegetation', and 'unproductive grass and shrubs', as defined by ref. 82. For each of these wooded pixels, we then predicted daily observation probabilities for the species distinguished by the Swiss forest vegetation database from February to November and averaged them. Then, we quantified taxonomic reporting bias for each species as the ratio between summed cover-abundance percentages across the surveys of the Swiss forest vegetation database and the corresponding summed observation probabilities, and corrected observation probabilities for all pixels and species by multiplying the raw observation probabilities with these bias estimates. Finally, for each pixel, we identified the potentially dominant species as the one with the highest bias-corrected observation probability. For the nationwide maps, potentially dominant species were aggregated to 200 m spatial resolution by mode. As a sensitivity analysis, we generated equivalent maps for the northern Ticino region based on the low-resolution NDCG DNN and SSDMs (Supplementary Fig. 3).

We validated predicted potentially dominant tree taxa against observed ones based on 1867 plots from the National Forest Inventory. As the coordinates of the NFI sites fell exactly on the corners of four pixels of our prediction layer, we extracted reweighted observation probabilities of all adjacent pixels and averaged them, before identifying the potentially dominant taxon. We evaluated prediction success for dominance predictions of both, raw observation probabilities, as well as for the bias-corrected observation probabilities, using top-1 accuracy to measure overall prediction success and $F_1$ score[113] to estimate performance for individual species.

**Projecting observation probabilities into the future.** In order to project observation probabilities to 2060, we first estimated future climatic conditions. To this end, we deduced expected deviations in temperature and precipitation for 2060 from the CH2018 projections of the National Centre for Climate Services[114]. In the region of the evaluated gradient, these projections estimated an increase in temperatures of about 2.5 °C and 1.75 °C for summer and winter respectively, a decrease in summer precipitation of about 17.5%, and an increase in winter precipitation of about 7.5% for the emission scenario RCP4.5. We updated the corresponding low-resolution variables with these changes, and used a univariate generalized additive model with thin-plate regression splines to estimate future frost change frequency as a function of future winter temperature. The other low-resolution predictors were assumed to remain constant. We then predicted observation probabilities of potentially dominant graminoid species under current and future conditions, for May first, June first, July first, and August first, and applied a taxonomic-reporting-bias correction as described above but using survey data from the dry meadows and pastures initiative. For plotting, we smoothed the bias-corrected curves of observation probability along the gradient, using local polynomial regression fitting[115] (R function loess) with a span of 0.15. As a sensitivity analysis, we also mapped bias-corrected habitat suitability based on SDM ensembles along the gradient under current conditions. Unless otherwise noted, model predictions were made in Python[89] (version 3.8.5), while subsequent analyses as well as mapping were done in the R environment (version 4.0.4), using the packages ROCR (version 1.0-11)[116], terra (version 1.7-71)[87], mgcv (version 1.8-42)[104], reticulate (version 1.30)[117], RColorBrewer (version 1.1-3)[118], and magick (version 2.7.4)[119].

### Reporting summary

Further information on research design is available in the Nature Portfolio Reporting Summary linked to this article.

## Data availability

Data sets used in this study include observational data and environmental data. The observational data from InfoFlora can be obtained for scientific research via online request under https://www.infoflora.ch/en/data/request-data.html. The accuracy of the spatial coordinates provided may be restricted depending on the context of the project and the confidentiality of the individual data. A standard extraction fee of CHF 175 may apply to user groups other than public research institutions. The observational data from the National Forest Inventory can be obtained after signing a written agreement, with contact details provided under https://www.lfi.ch/dienstleist/daten-en.php?lang=en. An extraction fee may apply to user groups other than public research institutions. The observational data from the Swiss Biodiversity Monitoring Program can be obtained after signing a written agreement with contact details provided under https://biodiversitymonitoring.ch/index.php/en/service/data-orders. Finally, the observational data from the Swiss forest vegetation database are available upon request as described under https://www.givd.info/ID/EU-CH-005. For environmental data, the LiDAR-based estimates of canopy height and inner forest density are available upon request as described under https://doi.org/10.16904/envidat.148; Landsat measurements of the normalized difference vegetation index (NDVI) are available under https://espa.cr.usgs.gov/; data from the EarthEnv-DEM90 digital elevation model can be downloaded und https://www.earthenv.org/DEM; reanalysis data on climate variables are available from https://doi.org/10.24381/cds.adbb2d47; plant indicator value-based estimates of soil properties, continentality, and light availability are available under https://doi.org/10.16904/envidat.153; annual and seasonal averages of high-resolution measurements of the enhanced vegetation index (EVI) for Switzerland are available under https://doi.org/10.26037/yareta:hapbjzl6dvbwnb5modewqozbfm, https://doi.org/10.26037/yareta:tilf3ibfnrafjpj6xpnea3vhpm, https://doi.

org/10.26037/yareta:of5ddowrxvbtjjurioduueopey, and https://doi.org/10.26037/yareta:hgw56omleveiplgftnd5ugwpja, for 2018, 2019, 2020, and 2021, respectively; high-resolution data on the vegetation height of Switzerland are available upon request as described under https://doi.org/10.16904/1000001.1; high-resolution data on forest type in Switzerland are available upon request as described under https://doi.org/10.16904/1000001.7; data on temperature and precipitation from the CHclim25 data set are available under https://doi.org/10.5281/zenodo.10635681; data on land use/land cover are available under https://doi.org/10.26037/yareta:dlx3hu54jfa3ne3c2xjfcnqpxm; high-resolution elevation data for Switzerland from the SwissAlti3D digital elevation model are available under https://www.swisstopo.admin.ch/en/height-model-swissalti3d; and high-resolution data on rivers and lakes from the SwissTLM3D topographic landscape model are available under https://www.swisstopo.admin.ch/en/landscape-model-swisstlm3d. Source data are provided with this paper.

## Code availability

Code underlying the main analyses conducted in this study, as well as minimal data sets to run it are available on GitHub (https://doi.org/10.5281/zenodo.10869585; Ref. 120).

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

## Acknowledgements

We thank all the experts and citizen scientists who contributed plant observations to the InfoFlora database, the Swiss Biodiversity Monitoring program, the Swiss forest vegetation database, and the dry meadows and pastures initiative, as well as Tobias Roth and for help with BDM data extraction. This study was supported by the Swiss Data Science Center (SDSC) grants no. c17-07 (SPEEDMIND) and c19-09 (COMECO) to N.E.Z., D.Z., P.D., D.N.K. and P.B. P.B., N.E.Z. & DNK acknowledge additional funding from the WSL-internal project PLAPP and D.N.K. & N.E.Z. acknowledge additional funding from the 2019-2020 BiodivERsA joint call for research proposal under the BiodivClim ERA-Net COFUND programme (project 'FeedBaCks') with the national funder Swiss National Science Foundation (grant no. 20BD21_193907).

## Author contributions

P.B. and N.E.Z. conceived the general idea and designed the study with the help of D.N.K., D.Z. and P.D. P.B. performed the main analysis. P.D. fitted the species distribution models. R.d.L. and J.D.W. supported the DNN implementation in the python environment. N.E.Z. and L.C.d.W. supported the ecological interpretation of the results. PB led the writing of the manuscript. All authors significantly interpreted results and contributed to writing and editing.

## Competing interests

The authors declare no competing interests.
