## [Peer Review File · Nature Communications]

Multispecies deep learning using citizen science data produces more informative plant community modelsREVIEWER COMMENTS

Reviewer #1 (Remarks to the Author):

The work proposed by the authors is extremely interesting. It allows to explore the potential of a new approach (i.e., multispecies DNNs) for finely analysing (1) the mapping variations in the phenology of selected forb species at the Switzerland scale; (2) the mapping dominant tree species; and (3) the projecting spatiotemporal distributions of graminoid species under severe climate change. All of this work has been initiated on a very large number of plant species, studied on a national scale and at a very fine spatial resolution. The quality of the work is outstanding, the writing is clear and concise, and the figures are rich and finely produced. Taken as a whole, this work seems to me to be suitable for the journal Nature Communication, and could make a significant contribution to the scientific community. I recommend acceptance of this manuscript, after consideration of the minor clarifications and corrections suggested below :

. Although the models developed were trained on data covering 2477 plant species and species aggregates, they were evaluated on 1339 species. I understand that the list of species on which the evaluation was carried out is related to the independent survey data sets available to the authors. The reader would benefit from a complementary information to better understand the (taxonomic, geographical, ecological) distribution of species that are not represented in the test set. As the training dataset is based on observations coming from citizen scientists, we might have thought that the independent datasets produced by the experts would cover a greater number of species (including rare species that are difficult for non-professionals to observe or determine). However, this does not appear to be the case. I think it would be useful to know a little more about the species present in the training and which could not be used for the evaluation, in order to strengthen the scientific impact of the conclusions obtained.

. Citizen science data offer an enormous opportunity for studying plant ecology and phenology. There are, however, strong biases that limit their direct use for some research questions. With regard to the phenology study carried out by the authors (sub-section "Predicting spatial variations in species' phenology"), the fact that a species is more easily

detectable and identifiable by non-specialists when a plant is in flowers can pose potential problems. It would therefore be useful to know what criteria the authors used to select the common forb species, for which the results are presented in Figure 2 and table 1. Biases could be assumed if, for example, the species selected all had the same morphological or ecological traits, which does not seem to be the case. However, a better justification of the choice of these species could reinforce the demonstration presented in this part of the manuscript.

. In the part of the article dedicated to the mapping of the timing of peak observation probability (tp_max), it might be relevant to briefly introduce in this part of the manuscript how this mapping was obtained through the use of the DNN. Then, in the paragraph "Timing of seasonal maxima in observation probability" (line 625), it would be useful to understand why a kernel size of 22 days was chosen, and to indicate whether the size of this kernel can have an influence on the estimated bias in the number of days, in table 1. These clarifications will make it easier for other research teams to exploit the method in other ecological and environmental contexts.

. With regard to the results presented in Table 1, do the authors have a hypothesis to explain the largest gap (i.e. 10 days) for the species *Orchis mascula*? Could elements relating to its ecology or detectability help to explain this?

. In their work on predicting potentially dominant tree species, the particularly low F1 scores for *Acer pseudoplatanus* (F1: 0.02) and *Fraxinus excelsior* (F1: 0.1), which have a quite large number of test observations, raise questions. What could explain such low scores and such large differences compared with the other species? What does an F1 score of 0 mean for *Betula pendula* (in Table 2)? Apart from a very small amount of training data, what could justify a score of zero?

. The results presented in Figure 4 are very original and interesting. However, the very low probability scores for most species in the RCP8.5 emission scenario (for the month of May in particular) raise questions about the robustness of the model in this context of use. The low number of species predicted with a score greater than 0.005 raises questions. Could the

authors put forward a hypothesis to explain this? The use of a complementary experiment, allowing to potentially see a gradual decline of these probabilities with the RCP8.5 scenario over a time period closer to the present (e.g. 2050 - 2070), would possibly allow a better understanding of this result.

. It would be useful in the sub-section dedicated to the description of the “network architecture and training settings” (line 551), to indicate the computational resources used / required for training these models. As the resources required may limit the reproducibility of this method or its replication in other larger contexts (such as larger areas, higher spatial resolutions, etc.), this information would be useful to share.

. The method that has been developed has a number of advantages, potentially extending far beyond the subjects on which it has been evaluated. Its use for estimating spatio-temporal variation in biodiversity and analyzing changes in habitat type could be mentioned among the potential prospects.

. Minors changes :

- The sentence in Line 60-61 : « For analyzing and biodiversity these biases ...” is not clear.

Can you try to clarify it?

- Uncomplete bibliographic reference (missing year of publication, etc.) : ”14 - Chauvier, Y. et al. Novel methods to correct for observer and sampling bias in presence-only species distribution models. Glob. Ecol. Biogeogr.”

Reviewer #2 (Remarks to the Author):

The paper presents a new multi-species distribution model based on a deep neural network trained on presence-only data (5M plant occurrences on the Swiss territory). Its originality compared to previous DeepSDMs is twofold: (i) the model is trained with a cost function minimizing the observed species rank rather than the cross-entropy loss and (ii), the model takes as input variable the day of year in addition to the other environmental covariates (bioclimatic, land use, etc.). Having the day of year as predictor allows to infer the species probability at any day of the year and to compute statistics from the resulting distribution.

In particular, it allows building a very high-resolution maps of the maximum probability of the timing of highest observation probability for a given species. It also allows averaging the probability of observation of the species along the year in order to have a potentially better estimator of the absence/presence probability of the species in a particular plot. The model is evaluated on both a subset of the presence-only data and a dataset of presence/absence data (not used for training the models). [1]

Overall, the paper is well written and the results obtained are original and interesting. The seasonal dimension, in particular, was not addressed in previous work about DeepSDM and is a clear and valuable contribution of the paper. The use of the model for the mapping of the most dominant tree species at the country level is also a very nice result.

Now, the second originality related to the use of the rank-based loss and its presumably supremacy over the cross-entropy loss is more questionable. In more details:

1- The paper states that the 'the cross entropy loss is not an adequate cost function for presence-only data' but this is not true. Indeed, the cross entropy loss is known to be a strictly proper loss [1], which means that it is minimized only when the model predicts the true conditional probability $p(y|x)$, i.e. the probability of the species given that a species has been observed in environment x . In the absence of taxonomic reporting bias, the cross-entropy is thus perfectly adapted to presence-only data. Asymptotically (with an infinite number of samples), it converges towards the true probability of observation of the species given the fact that one of them has been observed. So that contrary to what is said in the paper it does not "assume absences for all but the reported species". If N species are present in the same environment, the cross-entropy will asymptotically converges towards a probability of $1/N$ for each of them (again in the absence of taxonomic reporting bias). Note that it does not mean that a better loss function can not be found in the finite samples case. But it is wrong to say that the cross-entropy loss is not adapted to presence-only data. It has indeed strong theoretical guarantees of convergence in this regard.

2- On the other side, the theoretical justification of minimizing the rank of the observed species is unclear. If N species are present in the same environment (i.e. for a given x), it

might be meaningful to rank them according to ecological attributes such as dominance, coverage, abundance, etc. But since the proposed method is based on presence-only data, the predicted rank is not related to such attributes. It is more likely correlated with the taxonomic reporting bias. Indeed, among the N species present in a given environment x , the best guess of the species observed is the one that is observed the most often. Minimizing the rank might thus amplify the taxonomic reporting bias.

3- Concerning the comparative experiments of the two loss functions, there are also several questionable elements:

- a) Comparing the cross-entropy loss with the rank-based loss based on the rank of the predicted species itself is not really fair in the sense that only the second one is optimized towards this task. And as stated above, the task itself is questionable.
- b) The top- k accuracy metric is more relevant in this regard but since the used values for k are very low (1 and 5), it is also unlikely to well evaluate the ability of the models to predict the correct assemblage of present species. As for the rank-based metric, it rather estimates the ability of the model to predict the most observed species (the top-1 or top-5 most observed). So that, in the end, a lower top-1 accuracy for the cross-entropy might be a good sign that it is less sensitive to the taxonomic reporting bias. Thus, it would be necessary to also report the accuracy for larger values of k (ideally through a curve with growing k values). An interesting value for k , in particular, could be the average number of species present according to the presence/absence data.
- c) The objective and definition of the « weighted » version of those metrics is also unclear. Is the objective to give less importance to the most observed species or more importance ?? The article says « As we were interested in the performance of typical field observations, we calculated weighted means of these statistics, using taxon-wise numbers of training observations as weights ». As it is described, it lets suppose that it gives more importance to the most observed species which would be disconcerting because the most observed species are already strongly favored in the unweighted mean (because they have much more samples in the test set). If the authors mean that they use a weight inversely proportional to the taxon-wise numbers of training observations, this is fine. But the sentence should be rephrased and the objective should be explained more clearly.
- d) Concerning the AUC evaluation on presence/absence data, the way in which the habitat

suitability is estimated for each method is unclear. For the cross-entropy loss, in particular, it is unclear whether the suitability is estimated using the logit scores or the output of the softmax. This can make a big difference on the species-by-species AUC. For a fair comparison, it would be better to use the logit score since I guess this is at this layer of the network that the scores of the other model are extracted. And as stated in [2], the logit scores are more likely to represent the habitat suitability of each species separately than the categorical probabilities of the softmax.

4- Because of all those limitations, the paper in its current state is not convincing that the rank-based loss is a key ingredient towards training a good DeepSDM compared to the cross-entropy loss. In particular, the first sentence of the discussion saying that « our results highlight that when equipped with an appropriate cost function DNNs can efficiently learn the distributions of thousands of species » is not founded. It let the reader suppose that DNNs trained with cross-entropy loss cannot efficiently learn the distributions of thousands of species, which is incorrect (again the cross-entropy loss has some theoretical guaranty of convergence).

Overall, I believe the work is very valuable and original but a major revision should be achieved to be either more convincing (theoretically and experimentally) that the use of the rank-based loss is the central contribution of the paper, or to focus more the paper on the first contribution related to the seasonality of the prediction which is according to me very valuable.

Other minor remarks:

- the pooling of the seasonal predictions should be described more precisely. The sentence, « we approximated habitat suitability for each taxon and site as the 90th percentile of observation probability from April to September » is not clear enough. Do you compute a prediction for each day of year in the period ? And then take the 90th percentile in order to have a more robust estimator than taking the max ?

[1]Gneiting, T., & Raftery, A. E. (2007). Strictly proper scoring rules, prediction, and estimation. *Journal of the American statistical Association*, 102(477), 359-378.

[2] Deneu, B., Servajean, M., Bonnet, P., Botella, C., Munoz, F., & Joly, A. (2021). Convolutional neural networks improve species distribution modelling by capturing the spatial structure of the environment. *PLoS computational biology*, 17(4), e1008856.

Reviewer #3 (Remarks to the Author):

General comments

The authors propose and evaluate a new loss function for deep learning-based species distribution models (here abbreviated DNNs), and they explore various capabilities of these models that could bring new insights into their ecology: the spatial patterns of the peak flowering period, the spatial mapping of dominant species and their response to climate warming. There's a considerable experimental work and the idea of quantifying variations of phenology across space with these models for many species is great and new, and has potential. The authors illustrated the two capabilities and validated them with complementary standardized data (in the present period).

The detailed methodology is complex, but overall clearly described in the Methods and well synthesized in the introduction of the result. The method to correct taxonomic detection bias is simple and clever.

However, the methodology and results don't support some central results and I suggest below several sensitivity analysis and complementary experiments to validate the author claims :

- I'm not convinced so far about the claimed performance gain of the proposed rank-based loss function(NDCG) against the more standard cross-entropy (CE), as explained in my detailed comments below.

- Some methodological choices were surprising and unjustified while others could bias the

interpretation of the results. In models conception and learning, you introduce unjustified biases that could favour the high resolution model, but which are unrelated to the actual resolution of the input variables. For instance, the temporal sampling period of the data used to train the high res. model is more coherent with the test set, and you deprive a priori the low resolution model from potentially important information by removing certain input variables. In addition, the input variables are of different nature. See the point by points comments below. Therefore, you can't (and you don't) conclude that the input variable resolution explains the performance gain between the two NDCG implementations, but then I don't really see the point of keeping these 2 model versions. It seems to me that it would make more sense, be more synthetic (given the complex differences between input variables) to simply remove the low resolution models and directly compare the high res. NDCG to a comparable high res. CE model (learnt with the same species observation and input data).

-Besides, I also observe differences between the stacked SDM model and the DNNs (e.g. in the aggregation of taxons, in the training data) which I don't fully understand and that are not justified in the text, even though this is less important to the main results.

- When predicting into the future based on different climate (Section I.669), how do you guarantee that you are not extrapolating in many situations ? I don't see how your DNNs could generalize in extrapolation, i.e. when the input variable are far from their distribution in your training set. Maybe I missed something, but I think you should at least carry an analysis of prediction uncertainty, by diagnosing how far you are from training environmental conditions (e.g. conformal prediction) and model uncertainty, which is not straightforward with DNNs, but can be done with a DNNs ensemble.

Point by point comments :

Introduction

- 79-82 : What do you mean exactly? Maybe a reference would help to specify the idea?

Besides, pulling information from multiple species observations to correct for sampling bias

is not specific to DNN-based approaches, and can be done for instance a priori or in model, see e.g.

-Fithian, W., Elith, J., Hastie, T., & Keith, D. A. (2015). Bias correction in species distribution models: pooling survey and collection data for multiple species. *Methods in Ecology and Evolution*, 6(4), 424-438.

-Botella, C., Joly, A., Monestiez, P., Bonnet, P., & Munoz, F. (2020). Bias in presence-only niche models related to sampling effort and species niches: Lessons for background point selection. *PLoS One*, 15(5), e0232078.

- 79-82 : You seem to use interchangeably « spatial bias », « spatial sampling bias » and « sampling bias », which can lead to some confusion for the reader. As I understand it, sampling biases can include « spatial sampling bias » but also « taxonomic reporting bias » which you mention later. You seem to talk about « spatial sampling bias » in this sentence, so probably better to stick to this expression here.

- 90-94 : and more specifically « Like many common cost functions, the CEL therefore assumes absences for all but the reported species. » → This statement is not exactly true. The CEL just doesn't model absence probability, it models the relative probability across species CONDITIONALLY to the observation of a species. The problem is when the output relative probability across species (softmax), or any transform (e.g. its logit), is used improperly used as proxy of presence/absence probability or the relative abundance in space. The ranks of the species based on these outputs probabilities (when fitted on citizen science presence only data) represent how frequently a given species is observed compared to the others given some environmental conditions, which seems quite close to the aim of the loss you propose. Besides, the loss you propose (as explained in this paragraph), doesn't answer the presence/absence estimation problem either. Hence, I you should explain what exactly is lacking in the CEL that your loss should bring.

Results

- 131-132 : « We weighted the predicted ranks with the number of training observations per species to obtain scores that represent typical field observations » -> Do you mean that you computed the ratio of a species predicted rank multiplied by its number of training

observations, divided by the sum of this over species ? If so, i don't understand the rationale behind this as it would inflate the detection bias.

- 132-137 : « Observed species had significantly lower ranks in predictions of the low-resolution NDCG DNN than in predictions of the low-resolution CEL DNN and SSDMs, with weighted medians of 74, 76, and 175, respectively (Fig. 1a; $p \leq 0.001$ for all pairwise comparisons; $n = 12'187$). » → Could you please describe more explicitly your test (Null hypothesis, test statistic) At least in your response? This would deserve 1/2 sentence in your section « assessment of overall performance ». I had a look at your reference but it remained obscure to me. Besides, from Fig 1 a, it is VERY surprising that the predicted rank of the true species was significantly higher for « NDCG lowres. » than « CEL lowres. ». Given the huge sample size (>12,000), the p value might be sensitive to the test design. So I'm skeptical about this result.

Discussion

294-296 : At this stage, I'm not convinced about this result. The « NGCG lowres. » performance doesn't appear significantly better than the baseline approaches in 2 out of 3 evaluations in Figure 1 : In (a) it largely overlaps «CE lowres. » while in (b) it largely overlaps with « SSDM ». I couldn't assess the relevance of the statistical test.

315-316 : This correction makes sense intuitively, but you didn't evaluate it's impact on the predictive performance, which could be done, for instance, by comparing the F1 scores of Table 2 with and without this correction.

322-323 : « were confined to the relevant species » -> What do you mean? What type of species ?

332-336 : I'm not a specialist, but i'm not sure the realized phenology in this restricted spatio-temporal context can be interpreted as the potential phenological plasticity of the plant. In addition, as explained in the general comments, your model might be in extrapolation here, which is quite dangerous for deep learning models, and you didn't assess predictive uncertainty in these future conditions.

345-347 : Potentially, but this is not specific to rank-based multispecies DNN, is it ?

Methods

394-397 : «We sampled these test observations from the more recent version of the data set, wherever possible with observation dates after April 2019 (99.8% of cases) and where not possible with older observation dates» → It appears that this induces a positive bias in the evaluation of your NDCG high resolution model compared to the other models based on older data. You mostly evaluate models on data sampled after April 2019, a period included in the training data of the high resolution NDCG model (representing a large part of the training data, cf Extended Data Fig. 5), while the other models only used anterior data (given that your coarse resolution data dated from April 2019). Therefore, the data on which the high resolution NDCG model is fitted is more coherent with the test data than the data on which are fitted the coarse resolution models. This bias seems all the more likely as you didn't use a spatial block hold out procedure to isolate your test data points from your training data points, thus the high res. NDCG model can use the autocorrelation between training and test observations. Hence, I'm not convinced by the evaluation presented in Figure 1 a. For instance, degrade the spatial coordinates of the latest Info Flora extraction to fit the coarse resolution models.

515-520 : « For low-resolution DNNs, we sub-selected ten environmental predictors from the low-resolution set. The primary criteria thereby were variable importance in species distribution models, and absolute Pearson correlation coefficients of no more than 0.7 (Extended Data Fig. 6 and Extended Data Table 1). » -> Even though prior variable selection is common (and justified) practice in classic ecological/statistical settings (to avoid capturing spurious correlation etc), this practice is unusual when using deep neural networks because the variable selection generally happens automatically if your feature space (last hidden layers) have a reasonably small dimension. I imagine you do this to avoid capturing spurious correlations given your future prediction goal, but please justify.

520-521 : why doing prior variable selection in the low resolution DNN and not in the high resolution one ? Once again, this will induce an a bias in favour of the high res model

because it has access to more information in its input...

522-523 : « (2) our study area was heavily sampled » -> What do you mean ? More densely sampled than for the low res model ?

537-539 : How did you determine the training termination to control over-fitting? A common practice in deep learning is to split the data for learning into training and validation ($\leq 5\%$) and stop training when the validation loss/error metric start increasing (in trend).

540 : The explanation is nice but still a bit vague, why not inserting the loss formulas ?? The cross-entropy is a classic but I'm not familiar with the NDCG loss so would be great to have the formula at least for this one.

588-592 : Why doing so ? This is different than in the DNN case.

636-637 : Not sure here whether you refer to the same Info Flora data as the one you used to fit the model ? If so, this sounds more like a model fit metric. Plus, this evaluation doesn't correct for the effect spatio-temporal sampling patterns, does it?

656 : The procedure seems to make sense and is clever. However, the sentence is fully clear. You mean that you multiplied the predicted observation probability by the ratio of the previous sentence, right?

Detailed response to comments of Reviewer 1

R1.1:

The work proposed by the authors is extremely interesting. It allows to explore the potential of a new approach (i.e., multispecies DNNs) for finely analysing (1) the mapping variations in the phenology of selected forb species at the Switzerland scale; (2) the mapping dominant tree species; and (3) the projecting spatiotemporal distributions of graminoid species under severe climate change. All of this work has been initiated on a very large number of plant species, studied on a national scale and at a very fine spatial resolution. The quality of the work is outstanding, the writing is clear and concise, and the figures are rich and finely produced. Taken as a whole, this work seems to me to be suitable for the journal *Nature Communication*, and could make a significant contribution to the scientific community. I recommend acceptance of this manuscript, after consideration of the minor clarifications and corrections suggested below :

Thanks for this positive feedback. We are pleased that you judge the quality of the work as outstanding and share our enthusiasm about the potential of the approach.

R1.2:

. Although the models developed were trained on data covering 2477 plant species and species aggregates, they were evaluated on 1339 species. I understand that the list of species on which the evaluation was carried out is related to the independent survey data sets available to the authors. The reader would benefit from a complementary information to better understand the (taxonomic, geographical, ecological) distribution of species that are not represented in the test set. As the training dataset is based on observations coming from citizen scientists, we might have thought that the independent datasets produced by the experts would cover a greater number of species (including rare species that are difficult for non-professionals to observe or determine). However, this does not appear to be the case. I think it would be useful to know a little more about the species present in the training and which could not be used for the evaluation, in order to strengthen the scientific impact of the conclusions obtained.

Note that we used two datasets to validate the performance of multispecies DNNs (see subsection "Observations" under "Data" in the Methods). On the one hand, we had five test observations for each species from the citizen science data set, meaning that we did evaluate the model for 2465 of the 2477 modeled plant species modeled with DNNs, for which also SSDM predictions were available. On the other hand, we conducted an independent validation on expert-based survey data. This latter data set consists of a regular grid of ≈ 1500 surveys across Switzerland (see Extended Data Fig. 1). The reason why there are fewer species in this independent survey data are (1) that we have much fewer observations in the independent survey data, and (2) due to their regular distribution, the surveys mainly cover common habitats (agricultural land, common forest types, urban areas), whereas few surveys were made in rarer habitats with different communities (wetlands, dry meadows and pastures, remote alpine environments). The independent validation thus mainly considered common species, which we now emphasize in the main text (Lines 153-155).

R1.3:

. Citizen science data offer an enormous opportunity for studying plant ecology and phenology. There are, however, strong biases that limit their direct use for some research questions. With regard to the phenology study carried out by the authors (sub-section "Predicting spatial variations in species' phenology"), the fact that a species is more easily detectable and identifiable by non-specialists when a plant is in flowers can pose potential problems. It would therefore be useful to know what criteria the authors used to select the common forb species, for which the results are presented in Figure 2 and table 1. Biases could be assumed if, for example, the species selected all had the same

morphological or ecological traits, which does not seem to be the case. However, a better justification of the choice of these species could reinforce the demonstration presented in this part of the manuscript.

Actually, the fact that species are more easily detectable during the time they flower is exactly what we exploit for the phenological analysis. Unfortunately, however, this is not true to the same degree for all species, and the reviewer makes a valid point in demanding more transparency about the selection of the exemplary species. While it is a reasonable assumption for many species, e.g. for early-flowering geophytes like *Galanthus*, that they have an increased detectability during the time they flower, for canopy-forming trees, for example, this is usually not the case. Moreover, the relative likelihood of observing trees is increased during winter months, when few other species are easily detectable, which is one confounding effect in the approach. (We left out winter months for the phenological analysis to mask out problems arising from this effect). The selection of validated species was based on (a) ample data for model training ($\geq 10'000$) and (b) sufficient test observations with reported 'full bloom' phenology information for validation (more than ca. 30). From this subset, we picked a few examples with interesting spatiotemporal distribution patterns (Fig. 2). In order to be more transparent, we now show the scores of all species with $\geq 10'000$ training observations and ≥ 30 validation points in Extended Data Table 1, so that readers can see for themselves for which taxa the approach works well and for which it does not.

R1.4:

. In the part of the article dedicated to the mapping of the timing of peak observation probability (tp_max), it might be relevant to briefly introduce in this part of the manuscript how this mapping was obtained through the use of the DNN. Then, in the paragraph "Timing of seasonal maxima in observation probability" (line 625), it would be useful to understand why a kernel size of 22 days was chosen, and to indicate whether the size of this kernel can have an influence on the estimated bias in the number of days, in table 1. These clarifications will make it easier for other research teams to exploit the method in other ecological and environmental contexts.

We have now added a sentence summarizing how tp_max was identified in the main text (Lines 182-186). We used a moving-window approach to smooth the seasonal curves of observation probability. For each day, daily probabilities ranging from 11 days earlier to 11 days later were averaged. We chose 22 days (roughly three weeks) to have enough flexibility to identify maxima that represent seasonal peaks, but to be able to smooth out possible short-term fluctuations (at the level of few days). Usually, the direct model outputs of seasonal observation probabilities were rather smooth already, and thus kernel smoothing had little effect (see Fig. 2c for unsmoothed curves). However, for some rare combinations of environmental conditions or for marginal habitats, where the average observation probability for a species was comparably low, the seasonal curves were noisier and benefited more from such smoothing. Such smoothing could only introduce a small bias in the estimates of tp_max if the modes of seasonal observation probabilities were systematically skewed, i.e., if observation probabilities before to a mode would, e.g., increase more steeply than they decrease after the mode, but we have no theoretical and empirical reason to assume that this is the case. We have now introduced the tp_max approach in some more detail and added the aforementioned justification for the kernel size to the main text (Lines 695-699).

R1.5:

. With regard to the results presented in Table 1, do the authors have a hypothesis to explain the largest gap (i.e. 10 days) for the species *Orchis mascula*? Could elements relating to its ecology or detectability help to explain this?

Median observation probability of *Orchis mascula* peaked ten days (eight days in the revised MS with DNN ensemble-based predictions) before it was typically reported to be in 'full bloom'. One possible explanation for this may be that the species remains in an identifiable stage such as 'not yet flowered plant', 'budding', or 'early bloom' for a couple of days before it transitions to 'full bloom', and that citizen scientists start losing interest in the species (relative to others) during the second half of its 'full bloom'-phase. However, this is just a guess, and an in-depth analysis would be necessary for a well-informed explanation. Given the breadth of topics we cover in the manuscript, we believe investigating patterns at this level of detail is beyond the scope of our study.

Nevertheless, we now encourage the reader to look at Extended Data Table 1, where the scores for all phenology-validated species are listed, so that (s)he gets a complete picture about the temporal correspondence of *tp_max* and reported phenology states, which is not always as high as for the illustrated sample in the main manuscript (Lines 220-224).

R1.6:

. In their work on predicting potentially dominant tree species, the particularly low F1 scores for *Acer pseudoplatanus* (F1: 0.02) and *Fraxinus excelsior* (F1: 0.1), which have a quite large number of test observations, raise questions. What could explain such low scores and such large differences compared with the other species? What does an F1 score of 0 mean for *Betula pendula* (in Table 2)? Apart from a very small amount of training data, what could justify a score of zero?

Whether or not tree species dominate does not only depend on the environmental conditions but also on whether (a) forest management permits them to, (b) the forest is in a state of succession to which the respective species are specialized, and (c) whether species-specific demographic and dispersal processes allow the species to inhabit the location. As we tried to indicate in the discussion, predicting potential dominance becomes tricky for species that are widespread, but mainly dominate during early succession states (e.g., *Betula pendula*) or typically don't form large stands if they are not actively promoted (e.g., *Acer pseudoplatanus*). Such species only dominate in a small fraction of cells and the random expectation to find these few cells is close to zero (prevalence to the power of two). Accurately modelling such minority classes thus is already challenging, and the ecology of these species makes it even harder. It might be that additional predictors can improve the scores for some of the species, such as distance to forest edge for *Fraxinus excelsior* and especially more detailed remote sensing-based information. Future studies may explore to which extent this is possible. We have now discussed these points in more detail in the main text (Lines 354-359).

R1.7:

. The results presented in Figure 4 are very original and interesting. However, the very low probability scores for most species in the RCP8.5 emission scenario (for the month of May in particular) raise questions about the robustness of the model in this context of use. The low number of species predicted with a score greater than 0.005 raises questions. Could the authors put forward a hypothesis to explain this? The use of a complementary experiment, allowing to potentially see a gradual decline of these probabilities with the RCP8.5 scenario over a time period closer to the present (e.g. 2050 - 2070), would possibly allow a better understanding of this result.

Across all species, observation probabilities for a prediction to any point in space and time need to sum to one. The reweighting of observation probabilities to represent potential dominance may introduce some deviation here but having low probabilities for the potentially most dominant species indicates that the model is uncertain and distributes observation probabilities relatively flatly among species. This is not surprising, as projecting to the end of the century under a severe climate change scenario means heavy extrapolation as reviewer 3 correctly pointed out. We agree that the choice of period and emission scenario was a bit extreme for the proof of concept we attempted to make with

Fig. 4. We have now replaced Fig. 4 with a less severe scenario of climate change (RCP4.5) for 2060. Moreover, we have highlighted the sections of the gradient where the model extrapolates beyond training range to warn readers about uncertain projections. Projections of this less extreme scenario of climate change still show qualitatively similar patterns, but probability scores are generally higher, indicating less uncertainty in the model projections.

R1.8:

. It would be useful in the sub-section dedicated to the description of the “network architecture and training settings” (line 551), to indicate the computational resources used / required for training these models. As the resources required may limit the reproducibility of this method or its replication in other larger contexts (such as larger areas, higher spatial resolutions, etc.), this information would be useful to share.

We have now added the information that the training of the low-resolution models with the NDCG cost function takes about 76 hours and 13 hours with the CEL cost function, when a NVidia RTX 3090 GPU is used (subsection ‘Network architecture and training settings’).

R1.9:

. The method that has been developed has a number of advantages, potentially extending far beyond the subjects on which it has been evaluated. Its use for estimating spatio-temporal variation in biodiversity and analyzing changes in habitat type could be mentioned among the potential prospects.

Thank you for the hint. We have now added these prospects to the concluding paragraph.

R1.10:

. Minors changes :

- The sentence in Line 60-61 : « For analyzing and biodiversity these biases ...” is not clear. Can you try to clarify it?

Thanks for the hint. We have removed the “and”.

R1.11:

- Uncomplete bibliographic reference (missing year of publication, etc.) : “14 - Chauvier, Y. et al. Novel methods to correct for observer and sampling bias in presence-only species distribution models. Glob. Ecol. Biogeogr.”

We have complemented the missing information.

Detailed response to comments of Reviewer 2

R2.1:

The paper presents a new multi-species distribution model based on a deep neural network trained on presence-only data (5M plant occurrences on the Swiss territory). Its originality compared to previous DeepSDMs is twofold: (i) the model is trained with a cost function minimizing the observed species rank rather than the cross-entropy loss and (ii), the model takes as input variable the day of year in addition to the other environmental covariates (bioclimatic, land use, etc.). Having the day of year as predictor allows to infer the species probability at any day of the year and to compute statistics from the resulting distribution. In particular, it allows building a very high-resolution maps of the maximum probability of the timing of highest observation probability for a given species. It also allows averaging the probability of observation of the species along the year in order to have a potentially better estimator of the absence/presence probability of the species in a particular plot. The model is

evaluated on both a subset of the presence-only data and a dataset of presence/absence data (not used for training the models).

Overall, the paper is well written and the results obtained are original and interesting. The seasonal dimension, in particular, was not addressed in previous work about DeepSDM and is a clear and valuable contribution of the paper. The use of the model for the mapping of the most dominant tree species at the country level is also a very nice result.

Thank you for acknowledging the relevance and novelty of introducing seasonal resolution and mapping of potentially dominant tree species, which are two key results of the manuscript.

R2.2:

Now, the second originality related to the use of the rank-based loss and its presumably supremacy over the cross-entropy loss is more questionable. In more details:

In response to importance that reviewer 2 and 3 assigned to the results regarding model performance and cost functions, we have thoroughly reviewed the model configurations, which led us to the discovery that the hyperparameters for the CEL fit were not chosen well. In the original analysis we had chosen a lower batch size for the NDCG model because of memory constraints. Now we have ensured identical hyperparameters for NDCG and CEL models, which meant lowering the batch size for the CEL model from 100'000 to 250, and readjusted learning rates. After these changes, the performance of our CEL model is now considerably higher and very similar to that of the NDCG model (even slightly better with regards to predicting individual species distributions, see Fig. 1). Thank you for pointing us into this direction! We now find best results when the two model fits are ensembled (which was not the case before). We therefore kept the comparison of the two cost functions in the manuscript but now discuss their pros and cons more carefully (see below). Also, we reran the subsequent analyses with ensembled predictions, which generally led to better performance (except the performance of the predictions of potentially dominant tree species is slightly lower now). Moreover, we added the batch size setting to the methods (Line 586).

1- The paper states that the 'the cross entropy loss is not an adequate cost function for presence-only data' but this is not true. Indeed, the cross entropy loss is known to be a strictly proper loss [1], which means that it is minimized only when the model predicts the true conditional probability $p(y|x)$, i.e. the probability of the species given that a species has been observed in environment x . In the absence of taxonomic reporting bias, the cross-entropy is thus perfectly adapted to presence-only data. Assymptotically (with an infinite number of samples), it converges towards the true probability of observation of the species given the fact that one of them has been observed. So that contrary to what is said in the paper it does not "assume absences for all but the reported species". If N species are present in the same environment, the cross-entropy will assymptotically converges towards a probability of $1/N$ for each of them (again in the absence of taxonomic reporting bias). Note that it does not mean that a better loss function can not be found in the finite samples case. But it is wrong to say that the cross-entropy loss is not adapted to presence-only data. It has indeed strong theoretical garanties of convergence in this regard.

Reviewer 2 is right that we spent too few words and were too rough in explaining and motivating our choice of the NDCG cost function as relevant alternative to the cross-entropy loss, which we have corrected now.

By definition, the cross-entropy loss assumes mutually exclusive classes, i.e., it assumes that at every spatiotemporal location no more than one species is present, which clearly does not correspond to what we observe. It may be true that in the case of complete information and in the absence of

taxonomic reporting bias this limitation may be addressed assigning to each class present in a multi-label observation an individual single-label observation. But nevertheless, under real-world conditions with limited information, we have to expect that the CEL tends to reward the model for assuming that unobserved species are absent (or, more precisely, for allocating the full probability mass to species that have been observed under the given conditions).

We are not the first to point this out. The challenge of addressing a multi-label problem with single-label training data has been described before for both DNN-based species distribution modelling¹ and for applications in computer vision², and multiple cost functions have been shown to potentially perform better than the (binary) cross-entropy loss. We have now more carefully communicated these points in the introduction (Lines 87-104) and provide a detailed description of the two cost functions in the Methods (subsubsection “Cost functions”).

R2.3:

2- On the other side, the theoretical justification of minimizing the rank of the observed species is unclear. If N species are present in the same environment (i.e. for a given x), it might be meaningful to rank them according to ecological attributes such as dominance, coverage, abundance, etc. But since the proposed method is based on presence-only data, the predicted rank is not related to such attributes. It is more likely correlated with the taxonomic reporting bias. Indeed, among the N species present in a given environment x , the best guess of the species observed is the one that is observed the most often. Minimizing the rank might thus amplify the taxonomic reporting bias.

Modelling multispecies distributions based on presence-only data can be understood as finding the ID of a species that has been observed at a spatiotemporal location. And this, in turn, can be understood as a weak label ranking problem *sensu* Werner³ (2022). Although this perspective is somewhat uncommon in ecological modelling, such a model objective is meaningful and requires exactly the type of data we have at hand, and therefore, we believe, builds a stronger theoretical foundation than using a loss function that has been designed for a different problem (see R2.2). Moreover, while we agree that the NDCG cost function is susceptible to taxonomic reporting bias, we see no reason why this should be any more of a problem than for the CEL, and we disagree that the NDCG amplifies reporting bias. Our updated performance comparison confirms that for the data set at hand the CEL and the NDCG cost function perform similarly well (see Fig. 1, R2.5, R2.7 and R2.8). We have now more carefully communicated these points in the introduction (Lines 87-104) and provide more detailed description of the two cost functions in the Methods (subsubsection “Cost functions”).

R2.4:

3- Concerning the comparative experiments of the two loss functions, there are also several questionable elements:

a) Comparing the cross-entropy loss with the rank-based loss based on the rank of the predicted species itself is not really fair in the sense that only the second one is optimized towards this task. And as stated above, the task itself is questionable.

We believe that for the problem at hand ranks of the observed species in the predictions represent a very useful and intuitive performance metric, because they do not suffer from the limitation of top-1 and top-5 accuracy to only evaluate the predictions for the most observed species (as Reviewer 2 argues in R2.5) but provide threshold-independent overview over the relevance of *all* test predictions.

Moreover, it may be argued, in a similar fashion, that a performance comparison with top-1 accuracy is not fair because the cross-entropy loss assumes a presence of only one species at each spatiotemporal location, leading in particular to high top-1 accuracy, while this is less the case with the

NDCG cost function (this difference is evident in our updated performance results, see Extended Data Fig. 2). This is why we have used four different performance metrics to compare models trained with the two loss functions: median ranks, top-1 accuracy, top-5 accuracy, and AUC. In response to the reviewer's comments, we now also evaluate top-10 and top-20 accuracy (see R2.5, R2.7 and R2.8).

R2.5:

b) The top-k accuracy metric is more relevant in this regard but since the used values for k are very low (1 and 5), it is also unlikely to well evaluate the ability of the models to predict the correct assemblage of present species. As for the rank-based metric, it rather estimates the ability of the model to predict the most observed species (the top-1 or top-5 most observed). So that, in the end, a lower top-1 accuracy for the cross-entropy might be a good sign that it is less sensitive to the taxonomic reporting bias. Thus, it would be necessary to also report the accuracy for larger values of k (ideally through a curve with growing k values). An interesting value for k, in particular, could be the average number of species present according to the presence/absence data.

Thanks for this suggestion. We now also report model performance measured by top-10 and top-20 accuracy (see Extended Data Fig. 2).

R2.6:

c) The objective and definition of the « weighted » version of those metrics is also unclear. Is the objective to give less importance to the most observed species or more importance ?? The article says « As we were interested in the performance of typical field observations, we calculated weighted means of these statistics, using taxon-wise numbers of training observations as weights ». As it is described, it lets suppose that it gives more importance to the most observed species which would be disconcerting because the most observed species are already strongly favored in the unweighted mean (because they have much more samples in the test set). If the authors mean that they use a weight inversely proportional to the taxon-wise numbers of training observations, this is fine. But the sentence should be rephrased and the objective should be explained more clearly.

Thanks for pointing this out, the formulation was indeed confusing: The objective of the weighted version of those metrics was to give more importance to the most observed species. This was necessary, because our test set consisted of five observations per species and thus, in an unweighted fashion, our test results reflected taxonomically stratified scores. Given that reviewer 2 and 3 *a priori* expected performance scores to reflect the observed frequencies of species, we have now removed the introduction of weighting from the main text. We now describe weighting in the legend of Figure 1 and in the methods (Lines 666-669) and emphasize in the same sentence that our test set was taxonomically stratified.

2.7:

d) Concerning the AUC evaluation on presence/absence data, the way in which the habitat suitability is estimated for each method is unclear. For the cross-entropy loss, in particular, it is unclear whether the suitability is estimated using the logit scores or the output of the softmax. This can make a big difference on the species-by-species AUC. For a fair comparison, it would be better to use the logit score since I guess this is at this layer of the network that the scores of the other model are extracted. And as stated in [2], the logit scores are more likely to represent the habitat suitability of each species separately than the categorical probabilities of the softmax.

Thanks for this hint. We inferred AUC from the categorical probabilities of the softmax. In response to this comment, we have also evaluated AUC estimates based on the raw model outputs (before softmax). This, however, led to lower scores. Median species-by-species AUC was 0.937, 0.942, and 0.945 for the low-resolution NDCG DNN, the low-resolution CEL DNN, the low-resolution DNN

ensemble, respectively, when evaluated after softmax. When evaluated before softmax, the corresponding numbers were 0.796, 0.900, and 0.835. So, the scores for the CEL DNN were less affected than the scores for the other models, but still experienced a clear drop. Given these discouraging results, we did not add the comparison to the manuscript. Now, we explicitly state in the methods, that we evaluated AUC after softmax (Lines 678-679)

R2.8:

4- Because of all those limitations, the paper in its current state is not convincing that the rank-based loss is a key ingredient towards training a good DeepSDM compared to the cross-entropy loss. In particular, the first sentence of the discussion saying that « our results highlight that when equipped with an appropriate cost function DNNs can efficiently learn the distributions of thousands of species » is not founded. It let the reader suppose that DNNs trained with cross-entropy loss cannot efficiently learn the distributions of thousands of species, which is incorrect (again the cross-entropy loss has some theoretical guaranty of convergence).

This is a very relevant point, especially in light of the updated results. In agreement with these updated results, we have now adapted their interpretation from stating that the NDCG is advantageous over the CEL to stating that cost functions have different advantages and disadvantages (see *R2.2* and Lines 322-328).

R2.9:

Overall, I believe the work is very valuable and original but a major revision should be achieved to be either more convincing (theoretically and experimentally) that the use of the rank-based loss is the central contribution of the paper, or to focus more the paper on the first contribution related to the seasonality of the prediction which is according to me very valuable.

We thank reviewer 2 for pointing us towards a more careful comparison between the two loss functions. Based on these hints, we have now updated our results, and substantially revised introduction, discussion, and conclusions. We hope that these revisions now make a more convincing case.

R2.10:

Other minor remarks:

- the pooling of the seasonal predictions should be described more precisely. The sentence, « we approximated habitat suitability for each taxon and site as the 90th percentile of observation probability from April to September » is not clear enough. Do you compute a prediction for each day of year in the period ? And then take the 90th percentile in order to have a more robust estimator than taking the max ?

Yes, exactly. We now made the statement more clear (Lines 679-681).

[1]Gneiting, T., & Raftery, A. E. (2007). Strictly proper scoring rules, prediction, and estimation. *Journal of the American statistical Association*, 102(477), 359-378.

[2] Deneu, B., Servajean, M., Bonnet, P., Botella, C., Munoz, F., & Joly, A. (2021). Convolutional neural networks improve species distribution modelling by capturing the spatial structure of the environment. *PLoS computational biology*, 17(4), e1008856.

Detailed response to comments of Reviewer 3

R3.1:

General comments

The authors propose and evaluate a new loss function for deep learning-based species distribution models (here abbreviated DNNs), and they explore various capabilities of these models that could bring new insights into their ecology: the spatial patterns of the peak flowering period, the spatial mapping of dominant species and their response to climate warming. There's a considerable experimental work and the idea of quantifying variations of phenology across space with these models for many species is great and new, and has potential. The authors illustrated the two capabilities and validated them with complementary standardized data (in the present period).

The detailed methodology is complex, but overall clearly described in the Methods and well synthesized in the introduction of the result. The method to correct taxonomic detection bias is simple and clever.

Thank you for this positive feedback and for acknowledging the relevance and novelty of introducing seasonal resolution and mapping of potentially dominant tree species, which two key results of the manuscript.

R3.2:

However, the methodology and results don't support some central results and I suggest below several sensitivity analysis and complementary experiments to validate the author claims :

- I'm not convinced so far about the claimed performance gain of the proposed rank-based loss function(NDCG) against the more standard cross-entropy (CE), as explained in my detailed comments below.

Thanks for this important point. After revising the analysis, the results now indeed no longer show that NDCG cost function leads to higher performance than the CEL cost function (see *R2.2, R2.5, R2.7, R2.8*).

R3.3:

- Some methodological choices were surprising and unjustified while others could bias the interpretation of the results. In models conception and learning, you introduce unjustified biases that could favour the high resolution model, but which are unrelated to the actual resolution of the input variables. For instance, the temporal sampling period of the data used to train the high res. model is more coherent with the test set, and you deprive a priori the low resolution model from potentially important information by removing certain input variables. In addition, the input variables are of different nature. See the point by points comments below. Therefore, you can't (and you don't) conclude that the input variable resolution explains the performance gain between the two NDCG implementations, but then I don't really see the point of keeping these 2 model versions. It seems to me that it would make more sense, be more synthetic (given the complex differences between input variables) to simply remove the low resolution models and directly compare the high res. NDCG to a comparable high res. CE model (learnt with the same species observation and input data).

Note that in this study we did NOT aim to quantify the performance gain through high-resolution data. We agree with the reviewer that for such comparison the different predictor sets would be problematic and that, in principle, the two-resolution set up would not be necessary for what we aim to demonstrate here. The reason why we have set up the study at two resolutions is that, on one hand, state-of-the-art SDM maps were available at 100 m resolution, and on the other hand we wanted to explore the capacity of DNNs with the best information available (assuming better information leads to better model performance without aiming to find out exactly why and to what extent). SDM

projections were generated in 2019 based on a sophisticated modelling pipeline and requiring a substantial computational effort. Rerunning all SDMs at high resolution would thus be much more costly than running DNNs at two resolutions, and it would hardly affect our conclusions. Moreover, even if we did rerun the SDMs at high resolution we could not make a perfectly fair comparison, as for SDMs typically no more than ten predictors are recommended (especially for species with few observations and when models are used for projections)⁴, while with DNNs this complexity constraint is less limiting. We now include only low-resolution models for the Wilcoxon tests and mark the results from the high-resolution model in Fig. 1 in dashed lines, to emphasize that it was not part of the primary comparison. Moreover, rather than referring to the high-resolution model, we now refer to the model trained with the best information available in the 'Performance' subsection of the results.

As for the temporal period of the test observations, this has been done to assure that the test data for SDMs have not already been used for training (as no consistent train/test partitioning was done in their case). In other words, the choice of test data may not be perfectly fair to compare high- versus low-resolution DNNs (although the temporal mismatch is small), but it ensures a fair comparison between low-res models. We have now highlighted this in the methods (Lines 437-440).

R3.4:

-Besides, I also observe differences between the stacked SDM model and the DNNs (e.g. in the aggregation of taxons, in the training data) which I don't fully understand and that are not justified in the text, even though this is less important to the main results.

In the original version of the manuscript the training data for low-resolution DNNs and SDMs differed in three points: aggregation of taxons, different filtering steps, and overlap between training and some test data for SDMs. We have now rerun SDMs for species aggregates using the same taxonomic resolution as used in the DNNs (158 species aggregates affected), so the first type of difference is removed. With regards to filtering, we believe that this difference is justified: while for SDMs spatial thinning is required to reduce spatial sampling bias, for spatiotemporal DNNs observations without date information cannot be used. The third difference, that 0.2% of test observations have also been used for SDM training, we believe, is negligible, and would be advantageous for SDMs, which performed worse. We have now made these points clearer in the methods (Lines 441-444, 658-660).

R3.5:

- When predicting into the future based on different climate (Section I.669), how do you guarantee that you are not extrapolating in many situations? I don't see how your DNNs could generalize in extrapolation, i.e. when the input variable are far from their distribution in your training set. Maybe I missed something, but I think you should at least carry an analysis of prediction uncertainty, by diagnosing how far you are from training environmental conditions (e.g. conformal prediction) and model uncertainty, which is not straightforward with DNNs, but can be done with a DNNs ensemble.

This is a good point. We thought of Fig. 4 as a proof of concept that DNNs can be used for future projections of species distributions as SDMs are being used, with all the limitations that extrapolations with empirical models have. But the reviewer is right, we certainly make substantial extrapolations and did not highlight them, which may make an overconfident impression. Following R1.8, we have now switched to a less extreme scenario of climate change (to reduce extrapolation), and we highlight the sections of the gradient in which we extrapolated (grey-shaded area in Fig. 4e-h).

R3.6:

Point by point comments :

Introduction

- 79-82 : What do you mean exactly? Maybe a reference would help to specify the idea? Besides, pulling information from multiple species observations to correct for sampling bias is not specific to DNN-based approaches, and can be done for instance a priori or in model, see e.g. -Fithian, W., Elith, J., Hastie, T., & Keith, D. A. (2015). Bias correction in species distribution models: pooling survey and collection data for multiple species. *Methods in Ecology and Evolution*, 6(4), 424-438.

-Botella, C., Joly, A., Monestiez, P., Bonnet, P., & Munoz, F. (2020). Bias in presence-only niche models related to sampling effort and species niches: Lessons for background point selection. *PLoS One*, 15(5), e0232078.

Thank you for the references. We have now made the statement more specific and supported it with references.

R3.7:

- 79-82 : You seem to use interchangeably « spatial bias », « spatial sampling bias » and « sampling bias », which can lead to some confusion for the reader. As I understand it, sampling biases can include « spatial sampling bias » but also « taxonomic reporting bias » which you mention later. You seem to talk about « spatial sampling bias » in this sentence, so probably better to stick to this expression here.

Thanks for the suggested terms. We now avoid the term “sampling bias” and use either “spatial sampling bias” or “taxonomic reporting bias” throughout the manuscript.

R3.8:

- 90-94 : and more specifically « Like many common cost functions, the CEL therefore assumes absences for all but the reported species. » → This statement is not exactly true. The CEL just doesn't model absence probability, it models the relative probability across species **CONDITIONALLY** to the observation of a species. The problem is when the output relative probability across species (softmax), or any transform (e.g. its logit), is used improperly used as proxy of presence/absence probability or the relative abundance in space. The ranks of the species based on these outputs probabilities (when fitted on citizen science presence only data) represent how frequently a given species is observed compared to the others given some environmental conditions, which seems quite close to the aim of the loss you propose. Besides, the loss you propose (as explained in this paragraph), doesn't answer the presence/absence estimation problem either. Hence, I you should explain what exactly is lacking in the CEL that your loss should bring.

We agree that the CEL models conditional probabilities. For a detailed explanation of why we nevertheless argue that the CEL assumes absences for all but the reported species, see *R2.2*. Reviewer 3 is right that the CEL and the NDCG optimize a similar criterion, but the NDCG penalizes low probabilities for an observed class less strictly, if this class still ranks high in the model predictions. Consequently, the NDCG optimizes the model less strictly towards assuming absence for any species unless contrary evidence is available than the CEL does. By pushing for relative importance (low ranks), we may expect that the NDCG leads to model predictions with less peaked probability distributions. After the revision of the analysis, we now see these cost functions direct the model towards learning somewhat complementary, but similarly powerful rules, from which ensembling can benefit. We have now described these points in more detail in introduction (Lines 87-104), discussion (Lines 322-328), and methods (subsubsection “Cost functions”), including equations for the NDCG cost function.

R3.9:

Results

- 131-132 : « We weighted the predicted ranks with the number of training observations per species to obtain scores that represent typical field observations » -> Do you mean that you computed the ratio of a species predicted rank multiplied by its number of training observations, divided by the sum of this over species ? If so, i don't understand the rationale behind this as it would inflate the detection bias.

Yes, this is what we did. The reason was that the test set consisted of five observations per species, so it was, by default, taxonomically stratified. We realize that the description in its original version was confusing and have improved this now (see R2.6).

R3.10:

- 132-137 : « Observed species had significantly lower ranks in predictions of the low-resolution NDCG DNN than in predictions of the low-resolution CEL DNN and SSDMs, with weighted medians of 74, 76, and 175, respectively (Fig. 1a; $p \leq 0.001$ for all pairwise comparisons; $n = 12'187$). » → Could you please describe more explicitly your test (Null hypothesis, test statistic) At least in your response? This would deserve 1/2 sentence in your section « assessment of overall performance ». I had a look at your reference but it remained obscure to me. Besides, from Fig 1 a, it is VERY surprising that the predicted rank of the true species was significantly higher for « NDCG lowres. » than « CEL lowres. ». Given the huge sample size (>12,000), the p value might be sensitive to the test design. So I'm skeptical about this result.

We thank the reviewer for the hint, as we have not provided the full details here. We conducted paired, two-sided Wilcoxon signed rank tests between all pairwise comparisons of NDCG DNN, CEL DNN, DNN ensembles (in the revised MS), and SSDMs and corrected the resulting p -values for multiple comparisons using the Holm correction. Wilcoxon rank tests are standard non-parametric two-sample tests that are applied when the sample distributions do not meet the assumptions for a t -test.

We used the function `wilcox_test` of the R package `rstatix`⁵. This function calls the function `wilcox.test` (R base), which, in the paired case, tests the null hypothesis that the distribution of $x - y$ is symmetric around 0, where x and y are the two samples provided. The test-statistic thereby is based on positive and negative rank sums (see ref. ⁶ for more details). Given the paired design that we have here, we believe we can infer significant differences even if distributions largely overlap. There may be test observations that are harder to predict than others, and models may struggle similarly. But this is not the variation we are interested in. With a paired test we simply ask for each observation: which model did better? The differences may be subtle, but as long as they are systematic, we have a meaningful test result. We now make readers aware in the results that while the boxplots in Fig. 1 show distributions of scores across all observations, the Wilcoxon test tests for significant differences in score-differences per observation, i.e., the boxplots do not show the samples that are tested (Lines 134-137).

R3.11:

Discussion

294-296 : At this stage, I'm not convinced about this result. The « NGCG lowres. » performance doesn't appear significantly better than the baseline approaches in 2 out of 3 evaluations in Figure 1 : In (a) it largely overlaps «CE lowres. » while in (b) it largely overlaps with « SSDM ». I couldn't assess the relevance of the statistical test.

With the updated analysis we have now very similar performance for NDCG lowres. and CEL lowres., and mostly non-significant differences according to Wilcoxon tests. See above for the explanation why boxplots may largely overlap but Wilcoxon tests may still be significant.

R3.12:

315-316 : This correction makes sense intuitively, but you didn't evaluate it's impact on the predictive performance, which could be done, for instance, by comparing the F1 scores of Table 2 with and without this correction.

We now also show F1 scores without correction in Table 2 and compare the briefly in the corresponding section of the Results.

R3.13:

322-323 : « were confined to the relevant species » -> What do you mean? What type of species ?

We mean canopy-forming tree species. We have now rewritten the sentence to be clearer.

R3.14:

332-336 : I'm not a specialist, but i'm not sure the realized phenology in this restricted spatio-temporal context can be interpreted as the potential phenological plasticity of the plant. In addition, as explained in the general comments, your model might be in extrapolation here, which is quite dangerous for deep learning models, and you didn't assess predictive uncertainty in these future conditions.

Thank you for these comments, they are both important. With regards to the first point, it is true that if we assume that observation probability is linked to phenology, the observed realized potential phenology may underestimate the true plasticity if our study area does not capture the full phenological plasticity, much like the niche truncation issue that is known in species distribution modelling. We have now mentioned this limitation in the discussion (Lines 337-340).

With regards to the extrapolation, we also agree, and we have now reduced extrapolation and highlighted where it occurs (see *R3.5*).

R3.15:

345-347 : Potentially, but this is not specific to rank-based multispecies DNN, is it ?

We have adapted the statement.

R3.16:

Methods

394-397 : «We sampled these test observations from the more recent version of the data set, wherever possible with observation dates after April 2019 (99.8% of cases) and where not possible with older observation dates» → It appears that this induces a positive bias in the evaluation of your NDCG high resolution model compared to the other models based on older data. You mostly evaluate models on data sampled after April 2019, a period included in the training data of the high resolution NDCG model (representing a large part of the training data, cf Extended Data Fig. 5), while the other models only used anterior data (given that your coarse resolution data dated from April 2019). Therefore, the data on which the high resolution NDCG model is fitted is more coherent with the test data than the data on which are fitted the coarse resolution models. This bias seems all the more likely as you didn't use a spatial block hold out procedure to isolate your test data points from your training data points, thus the high res. NDCG model can use the autocorrelation between training and test

observations. Hence, i'm not convinced by the evaluation presented in Figure 1 a. For instance, degrade the spatial coordinates of the latest Info Flora extraction to fit the coarse resolution models.

In this study we did NOT aim to compare effects of resolution on model performance. Rather, we worked at two resolutions to have a fair comparison between SSDMs, NDCG DNNs and CEL DNNs on one side (low resolution), and the best available information for the phenological analysis and the analysis of potential dominance on the other side (see R3.3 for more information on this point).

R3.17:

515-520 : « For low-resolution DNNs, we sub-selected ten environmental predictors from the low-resolution set. The primary criteria thereby were variable importance in species distribution models, and absolute Pearson correlation coefficients of no more than 0.7 (Extended Data Fig. 6 and Extended Data Table 1). » -> Even though prior variable selection is comon (and justified) practice in classic ecological/satistical settings (to avoid capturing spurious correlation etc), this practice is unusual when using deep neural networks because the variable selection generally happens automatically if your feature space (last hidden layers) have a reasonably small dimension. I imagine you do this to avoid capturing spurious correlations given your future prediction goal, but please justify.

We did this to have (1) roughly the same number of predictors as were available to SDMs, and, as Reviewer 3 assumed, (2) to avoid having a too complex model for future projections. We have now added these justifications to the methods.

R3.18:

520-521 : why doing prior variable selection in the low resolution DNN and not in the high resolution one ? Once again, this will induce an a bias in favour of the high res model because it has access to more information in its input...

That's correct. But our aim was never to test the effect of resolution on model performance (see R3.3 and R3.16).

R3.19:

522-523 : « (2) our study area was heavily sampled » -> What do you mean ? More densely sampled than for the low res model ?

No, we meant that given the number of data points we do not expect big holes in the modeled environmental space and thus not much extrapolation when mapping phenology and potentially dominant species under current conditions. This is a minor point and we have now removed it to avoid confusions.

R3.20:

537-539 : How did you determine the training termination to control over-fitting? A comon practice in deep learning is to split the data for learning into training and validation (<=5%) and stop training when the validation loss/error metric start increasing (in trend).

Thank you for this hint. We have now created a validation set of the same size as the test set and deduced training termination in the way reviewer 3 suggests from a preliminary validation run with the low-resolution models (see Supplementary Information).

R3.21:

540 : The explanation is nice but still a bit vague, why not inserting the loss formulas ?? The cross-

entropy is a classic but I'm not familiar with the NDCG loss so would be great to have the formula at least for this one.

We have now rewritten this section, adding more details and formulas.

R3.22:

588-592 : Why doing so ? This is different than in the DNN case.

The SDMs maps had already been generated in 2019 with this different taxonomic set up. In order to make the results as comparable as possible, we have now rerun SDMs for species aggregates, using the same approach as for DNNs (see also *R3.4*).

R3.23:

636-637 : Not sure here whether you refer to the same Info Flora data as the one you used to fit the model ? If so, this sounds more like a model fit metric. Plus, this evaluation doesn't correct for the effect spatio-temporal sampling patterns, does it?

It is true that the test set for phenology is a subset of the Info Flora training (and test) data, but the validations are made for an attribute of the observations that was not seen by the model. Moreover, as we explain in *R1.3*, we only evaluated phenological predictions for species with >10'000 observations, and those observations with phenology information of the category 'full bloom' typically only made up a small fraction of the total number observations (median = 0.5% per species). Most observations do not contain phenological information, and those who do may report several other phenological stages, too. Given the small impact of these minor fractions on the model fit, and the fact that phenological patterns fluctuate substantially from year to year, we do not believe that the metrics are just goodness of fit measures. We now inform the readers about this potential problem (Lines 716-720), and we have added the total number of training observations and the number of phenology observations to Extended Data Table 1 so that readers can judge for themselves.

R3.24:

656 : The procedure seems to make sense and is clever. However, the sentence is fully clear. You mean that you multiplied the predicted observation probability by the ratio of the previous sentence, right?

This is correct, thanks. We have clarified the sentence.

Bibliography

1. Aodha, O. Mac, Cole, E. & Perona, P. Presence-Only Geographical Priors for Fine-Grained Image Classification. in *2019 IEEE/CVF International Conference on Computer Vision (ICCV)* 9595–9605 (IEEE, 2019). doi:10.1109/ICCV.2019.00969.
2. Cole, E. *et al.* Multi-Label Learning from Single Positive Labels. (2021).
3. Werner, T. A review on instance ranking problems in statistical learning. *Mach. Learn.* **111**, 415–463 (2022).
4. Brun, P. *et al.* Model complexity affects species distribution projections under climate change. *J. Biogeogr.* **47**, 130–142 (2020).
5. Kassambara, A. rstatix: Pipe-Friendly Framework for Basic Statistical Tests. at <https://cran.r-project.org/package=rstatix> (2023).
6. Hollander, M., A. Wolfe, D. & Chicken, E. *Nonparametric Statistical Methods*. (Wiley, 2015). doi:10.1002/9781119196037.

REVIEWER COMMENTS

Reviewer #1 (Remarks to the Author):

The authors have done a great work to clarify the points previously raised, and have significantly improved the quality of the explanations required for a proper understanding of their work.

In particular, they have better justified and described the two test sets used to evaluate the performance of their multispecies model. The availability of scores for all the species with $\geq 10,000$ training observations and ≥ 30 validation points, in the "Extended Data Table 1", meets the need I had raised to share more detailed information by species, for the selection of the validated species.

The information provided for a better description of the timing of peak observation probability (tp_max) seems to me to be convincing and sufficient for a good understanding of the approach implemented by the authors.

The explanations provided by the authors in their response letter and in the manuscript, for the moderate performance of some of their results on the species that I raised in my previous review (*Orchis mascula* and *Acer pseudoplatanus*), seem to be satisfactory. I would particularly like to thank the authors for the precautions taken in their assumptions to explain the limited performances in the case of predictions of potentially dominant tree species.

The modification of Figure 4 with a less extreme climate scenario that is closer in time, is more likely to enable the reader to better appreciate the gap between current and future conditions that can be observed thanks to the authors' methodology.

The changes made in response to the comments made by the two other reviewers (in particular the addition of an Ensemble model, and the clarification of the cost function used) make it possible to share the results obtained with a more in-depth analysis, in particular to convince the scientific community of the relevance of the methodology developed.

As a result of all the improvements made and the quality and originality of the results obtained, I feel that the proposed manuscript is now suitable for publication.

Reviewer #2 (Remarks to the Author):

The authors carefully addressed all my concerns and recommendations. I think the paper is now in a state acceptable for publication.

Reviewer #3 (Remarks to the Author):

Thanks for clarifying many things in the revised MS and response letter. Overall, the methodology appears clearer and better justified. Specifically, your validation scheme for the prediction of the day of peak flowering makes now sense to me.

My main comments are about the positioning and hierarchy of your results, and I invite you to adapt it:

The claimed contribution of showing the predictive performance of multi-species DNN over stacked SDM is not new, so it should rather be formulated as a consolidation of existing results. Indeed, it has been already shown and published several times, as developed in a comment below.

The result that appears original and the most convincing to me is the prediction of the phenological peak day. Also, the associated perspectives for ecological research mentioned in discussion are very interesting.

The correction of taxonomic sampling bias appears nice in principle, but it only improves the prediction for only 3 out of 13 species. Hence, it's not really convincing, but not explained nor discussed.

The potential of deep learning for future species predictions of phenology or dominance under climate change is difficult to assess as there is no validation nor comparison with other a simpler predictive approach. As argued in a comment below, deep learning is risky for "out-of-sample" prediction and could lead to largely erroneous conclusions if applied widely and blindly.

Based on what evidence should people trust future predictions of this approach in light of

the known limitations of deep learning?

Specifics comments:

37-38: "with unprecedented 37 efficiency." -> No, you didn't show that as you didn't compare your DNN to past multi-species DNNs trained on citizen science data, like

Chen 2016, Deneu , Cole, Estopinan, Kellenberg etc

Besides, if you rather refer to the fact that DNN outperform more classical SDM, this is not new.

it has been already shown for plants on several large scale datasets

79-81: Not sure I get this point. Why modeling species jointly per se would reduce sampling biases? It's not wrong but the link is not made clearly.

In Fithian & Hastie, the disentanglement of sampling effort vs species densities comes from three (i) the sampling effort model -> a common multiplicative function to the occurrence rate of all modelled species, (ii) the integration of PA (not affected by sampling bias) and (iii) the information gained on sampling effort through all species occurrences through model (i). Hence, it's not only the joint modeling of multiple species that allows to correct sampling bias, but also the model design.

92-93: "Yet, the CEL assumes classes to be mutually exclusive, and thus exactly one species to be present per observation. Presence-only observations, however, are single positive multi-label data, meaning that we expect multiple species to be present per observation, but only have information about the presence of one of them (and no information about the presence or absence of all other species)."

Not exactly, indeed the CEL assumes that one class is associated to one observation, but many observations may be associated to one point (in geographic or environmental space) so that many species may be observed at that point. The situation actually arises often with massive citizen science dataset.

In that case, the CEL is minimized for that point when the predicted species probabilities

reflect the proportions of observations of each species among all observations at the point. Hence, in the absence of sampling biases and under independent and dense sampling events, the species probabilities obtained with the CEL reflect the community composition.

Fig 3c: I'm surprised that the dominance is alternated at high spatial resolution between *Castanea sativa* and *Quercus pubescens*, given that the former is characteristic of acid soils while the latter characteristic of higher Ph / alkaline soils. Do these dominance patterns actually reflect changes in soil pH?

I'm not used to see Ph varying so much at such a high spatial resolution.

272-273: How did you binarized the bias-corrected observation probability to compute the F1 scores? Did you use some threshold or did you take the threshold maximizing the F1 per species? I didn't find it in Mat & met either.

Table 2: how do you interpret that only 3 out of 13 species benefited from taxonomic sampling bias correction?

Discussion

318-321: The first statement is vague and the corresponding scientific contribution itself isn't new. What does "Efficiently" mean? This is not a factual way to summarize scientific results.

The following statement "DNN were better than stacked sdm" is more factual. Still, you only compared DNN approaches to a specific implementation of stacked SDMs, while there are tons of types of approaches for single species SDMs out there, questioning the robustness of this result.

Besides, the fact that multi-species DNN outperform classic stacked single species SDM was already shown several times, for instance by Chen et al., 2017

(<https://arxiv.org/abs/1609.09353>) for birds and Botella et al., 2018 (that you cited) for plants.

Note that comprehensive SDM assessments comparing many methods (DNNs, CNNs, stacked SDMs based on Random forests, boosted trees, Maxent, etc) were carried during

the six past editions of GeoLifeCLEFs, an annual open SDM evaluation campaign, as illustrated by the campaigns' overviews (2022: <file:///C:/Users/user/Downloads/paper-155.pdf>, 2023: <https://ceur-ws.org/Vol-3497/paper-166.pdf>).

You could instead formulate that your results consolidate the performance of DNN on a new dataset.

346-366: Could you explain or discuss why the taxonomic sampling bias correction decreased the performance for 10 out of 13 species? It doesn't seem related to the commonness of the species from Table 2.

378-380: Most likely, but why are DNN the best tool to do that? Don't you think there's a greater risk of low temporal transferability with DNN predictions when future changes in climate or land use disrupt the joint distribution of input environmental variables? DNNs have no guarantee of robustness for this kind of extrapolation (unlike models integrating general constraints of ecological niches), I don't know of any regularisation mechanism that prevents this, and quantifying the uncertainty of point predictions is a notorious limit of deep learning models.

Mat & met:

613: In equation (1), what mean "n" then? Is it the rank up to which the loss is evaluated as suggest in rows 23-24? Please clarify.

727-29: Did you test alternative strategies than averaging over days, like taking the maximum? Averaging might under-predict species with shorter blooming period, no?

Detailed response to comments of Reviewer 1

R1.1:

The authors have done a great work to clarify the points previously raised, and have significantly improved the quality of the explanations required for a proper understanding of their work.

In particular, they have better justified and described the two test sets used to evaluate the performance of their multispecies model. The availability of scores for all the species with $\geq 10,000$ training observations and ≥ 30 validation points, in the "Extended Data Table 1", meets the need I had raised to share more detailed information by species, for the selection of the validated species.

The information provided for a better description of the timing of peak observation probability (tp_max) seems to me to be convincing and sufficient for a good understanding of the approach implemented by the authors.

The explanations provided by the authors in their response letter and in the manuscript, for the moderate performance of some of their results on the species that I raised in my previous review (*Orchis mascula* and *Acer pseudoplatanus*), seem to be satisfactory. I would particularly like to thank the authors for the precautions taken in their assumptions to explain the limited performances in the case of predictions of potentially dominant tree species.

The modification of Figure 4 with a less extreme climate scenario that is closer in time, is more likely to enable the reader to better appreciate the gap between current and future conditions that can be observed thanks to the authors' methodology.

The changes made in response to the comments made by the two other reviewers (in particular the addition of an Ensemble model, and the clarification of the cost function used) make it possible to share the results obtained with a more in-depth analysis, in particular to convince the scientific community of the relevance of the methodology developed.

As a result of all the improvements made and the quality and originality of the results obtained, I feel that the proposed manuscript is now suitable for publication.

Thank you for this positive feedback. We are pleased that you are satisfied with our effort to clarify our methodological set up and to transparently report the strengths and weaknesses of our approach.

Detailed response to comments of Reviewer 2

R2.1:

The authors carefully addressed all my concerns and recommendations. I think the paper is now in a state acceptable for publication.

We are happy that our revisions could satisfy the relevant concerns raised in the previous round. Thank you for this positive feedback.

Detailed response to comments of Reviewer 3

R3.1:

Thanks for clarifying many things in the revised MS and response letter. Overall, the methodology appears clearer and better justified. Specifically, your validation scheme for the prediction of the day of peak flowering makes now sense to me.

My main comments are about the positioning and hierarchy of your results, and I invite you to adapt it:

The claimed contribution of showing the predictive performance of multi-species DNN over stacked SDM is not new, so it should rather be formulated as a consolidation of existing results. Indeed, it has been already shown and published several times, as developed in a comment below.

Thank you for taking the time for another round of in-depth feedback. This is a fair point; we have not done sufficient justice to previous work comparing DNNs with SDMs. We have now brought this context into the discussion and clarified that there have been similar comparisons before ours (Lines 328-332).

R3.2:

The result that appears original and the most convincing to me is the prediction of the phenological peak day. Also, the associated perspectives for ecological research mentioned in discussion are very interesting.

Thanks for acknowledging the relevance of the phenological perspective. We agree that this is an important perspective to keep investigating in future work and has immediate relevance for many applications.

R3.3:

The correction of taxonomic sampling bias appears nice in principle, but it only improves the prediction for only 3 out of 13 species. Hence, it's not really convincing, but not explained nor discussed.

We find this a fairly pessimistic way of interpreting our results. The bias correction was primarily intended to improve the overall accuracy, and it did so from 0.38 to 0.50, so by clearly over 30%. One could argue that many (if not most) benchmarking studies in computer science report improvements that are more modest. Still, we agree with the reviewer that the fact that F1 scores did decrease for most species deserves some discussion. Our explanation of this result is that bias-correction particularly increased true positives of beech and spruce, the two species that dominate the majority of Swiss forest stands (with 352 additional true positives), while true positives for silver fir and larch decreased by 135. The prevalences of the remaining nine species were below 5% and changes in true positives were negligible. At a closer look, the improvement in overall accuracy from bias correction is almost entirely owed to the upweighting of spruce, while for beech the increase in false positives roughly offset increase in true positives and the decrease

in false negatives (F1 score remained stable). This potentially happened at the cost of losing true positives of silver fir and indicates that the bias correction estimates we obtained from the Swiss forest vegetation database may not be perfect for the test set. While bias-correction is clearly beneficial for overall accuracy, we agree that for research questions focusing on rarer species, uncorrected predictions may be more relevant. We have added these considerations to the discussion (Lines 370-376). Moreover, we now added the bias correction weights to Table 2, for readers to have a better understanding of their effect.

R3.4:

The potential of deep learning for future species predictions of phenology or dominance under climate change is difficult to assess as there is no validation nor comparison with other a simpler predictive approach. As argued in a comment below, deep learning is risky for "out-of-sample" prediction and could lead to largely erroneous conclusions if applied widely and blindly. Based on what evidence should people trust future predictions of this approach in light of the known limitations of deep learning?

Thank you for this comment. As we tried to point out in the last round of revisions, our explorations of potential future changes in spatiotemporal distribution are not supposed to be a blueprint for DNN-based model extrapolation but rather an illustration of how phenology and potential dominance can offer new insight on distributional change. We agree that we had not addressed this point clearly in the discussion. We fully agree that out-of-sample predictions are problematic and should be avoided as much as possible. Still, there is a high interest in estimates of climate change impact on species distributions, for example for climate-smart biodiversity conservation and there are established ways of limiting extrapolation. Nested approaches can be used, for example, to train climate niches at the continental scale, and ideally cover entire species' distributions, thus avoiding niche truncation¹. Moreover, there are also developments for DNNs to inform unavoidable extrapolations by local trends at the edges of the training range (see ref. ² for an example of regression with neural networks). Finally, the approach we proposed may also be relevant for investigations on past distributional change. We have now more clearly emphasized these points in the discussion, and the fact that more research is needed for a thorough validation of the ideas (Lines 395-407).

R3.5:

Specifics comments:

37-38: "with unprecedented 37 efficiency." -> No, you didn't show that as you didn't compare your DNN to past multi-species DNNs trained on citizen science data, like

Chen 2016, Deneu , Cole, Estopinan, Kellenberg etc

Besides, if you rather refer to the fact that DNN outperform more classical SDM, this is not new. it has been already shown for plants on several large scale datasets

We have rewritten the sentence, focusing on the degree of detail we can provide rather than the novelty of DNNs. With regards to comparing SDMs to DNNs, see *R3.1*.

R3.6:

79-81: Not sure I get this point. Why modeling species jointly per se would reduce sampling biases? It's not wrong but the link is not made clearly.

In Fithian & Hastie, the disentanglement of sampling effort vs species densities comes from three (i) the sampling effort model -> a common multiplicative function to the occurrence rate of all modelled species, (ii) the integration of PA (not affected by sampling bias) and (iii) the information gained on sampling effort through all species occurrences through model (i).

Hence, it's not only the joint modeling of multiple species that allows to correct sampling bias, but also the model design.

Thank you for this explanation, with which we agree. We have now added a sentence specifying that – if we assume spatial sampling bias to be similar across species – we assume it to have a lower effect on conditional probabilities of multispecies models than on habitat suitability scores of individual SDMs that are derived from contrasting presence observations against random background points (Lines 82-86). Note that although there are ways to mimic sampling bias in the pseudoabsences of SDMs, such as the target group approach³, their advantage over random background points mainly exists for heavily-biased, sparse data⁴ and much less for citizen science observations such as that of Swiss plant species⁵ that are massive compared to the area covered.

R3.7:

92-93: "Yet, the CEL assumes classes to be mutually exclusive, and thus exactly one species to be present per observation. Presence-only observations, however, are single positive multi-label data, meaning that we expect multiple species to be present per observation, but only have information about the presence of one of them (and no information about the presence or absence of all other species)."

Not exactly, indeed the CEL assumes that one class is associated to one observation, but many observations may be associated to one point (in geographic or environmental space) so that many species may be observed at that point. The situation actually arises often with massive citizen science dataset.

In that case, the CEL is minimized for that point when the predicted species probabilities reflect the proportions of observations of each species among all observations at the point. Hence, in the absence of sampling biases and under independent and dense sampling events, the species probabilities obtained with the CEL reflect the community composition.

Agreed. A similar point has been raised by Reviewer 2 in the last round. The problem is that we cannot expect "absence of sampling biases and [...] independent and dense sampling events". Nevertheless, for some groups, such as orchids, our data was quite dense, which might have been one reason for the comparably high performance of the CEL. For other groups, such as ferns, however, the sampling was much more patchy, and model performance worse. We have added the possibility that multiple observations may be made for the same spatiotemporal location (Lines 98-99).

R3.8:

Fig 3c: I'm surprised that the dominance is alternated at high spatial resolution between *Castanea sativa* and *Quercus pubescens*, given that the former is characteristic of acid soils while the latter characteristic of higher Ph / alkaline soils. Do these dominance patterns actually reflect changes in soil pH?

I'm not used to see Ph varying so much at such a high spatial resolution.

From an inspection of the pH map it seems that the region in Fig. 3c is moderately acidic and that the small-scale variation between dominance of *Quercus pubescences* and *Castanea sativa* is rather related to exposition. The depicted region is on loose rock with crystalline bedrock. Flora Helvetica states that *Castanea sativa* typically grows on soils with a pH ranging from 3.5 to 6.5 while the corresponding range for *Quercus pubescences* is from 4.5 to 7.5. It seems, thus, as if soil acidity is not limiting either species in the depicted region.

R3.9:

272-273: How did you binarized the bias-corrected observation probability to compute the F1 scores? Did you use some threshold or did you take the threshold maximizing the F1 per species? I didn't find it in Mat & met either.

Thanks for the hint. We now added a sentence of explanation (Lines 273-276). We did not calculate F1 from presence/absence of all species but from potential dominance, so we made only one comparison per site. After bias-correction, we identified the species with the highest reweighted observation probability as potentially dominant and compared it to the species with the most observed individuals with ≥ 12 cm stem diameter at breast height.

R3.10:

Table 2: how do you interpret that only 3 out of 13 species benefited from taxonomic sampling bias correction?

See R3.3.

R3.11:

Discussion

318-321: The first statement is vague and the corresponding scientific contribution itself isn't new. What does "Efficiently" mean? This is not a factual way to summarize scientific results. The following statement "DNN were better than stacked sdm" is more factual. Still, you only compared DNN approaches to a specific implementation of stacked SDMs, while there are tons of types of approaches for single species SDMs out there, questioning the robustness of this result.

Besides, the fact that multi-species DNN outperform classic stacked single species SDM was already shown several times, for instance by Chen et al., 2017 (<https://arxiv.org/abs/1609.09353>) for birds and Botella et al., 2018 (that you cited) for plants. Note that comprehensive SDM assessments comparing many methods (DNNs, CNNs, stacked

SDMs based on Random forests, boosted trees, Maxent, etc) were carried during the six past editions of GeoLifeCLEFs, an annual open SDM evaluation campaign, as illustrated by the campaigns' overviews (2022: <file:///C:/Users/user/Downloads/paper-155.pdf>, 2023: <https://ceur-ws.org/Vol-3497/paper-166.pdf>).

You could instead formulate that your results consolidate the performance of DNN on a new dataset.

Thanks for these references. We have now rephrased the corresponding section of the discussion according to these suggestions (Lines 328-332).

R3.12:

346-366: Could you explain or discuss why the taxonomic sampling bias correction decreased the performance for 10 out of 13 species? It doesn't seem related to the commonness of the species from Table 2.

Done (Lines 370-376). See also *R3.3*. Change in F1 may not be related to commonness but true positives clearly are, with 352 additional true positives for beech and spruce, and 103 less for silver fir, whereas the changes for all other species are minor.

R3.13:

378-380: Most likely, but why are DNN the best tool to do that? Don't you think there's a greater risk of low temporal transferability with DNN predictions when future changes in climate or land use disrupt the joint distribution of input environmental variables? DNNs have no guarantee of robustness for this kind of extrapolation (unlike models integrating general constraints of ecological niches), I don't know of any regularisation mechanism that prevents this, and quantifying the uncertainty of point predictions is a notorious limit of deep learning models.

Agreed. As argued above our goal here was highlighting that phenology and potential dominance are essential components of distributional change that are often forgotten, rather than offering a DNN-based blueprint for extrapolatory analyses. As argued above, we agree that such extrapolations can be problematic, and we would further argue that the case is not fundamentally different for algorithms like random forest that are more commonly used in the SDM literature. Even simple algorithms like GLMs do not necessarily set realistic constraints to ecological niches, unless the training data covers the entire niche of a species of interest (i.e., no niche truncation in the data). We have now added these points to the discussion (Lines 395-407).

R3.14:

Mat & met:

613: In equation (1), what mean "n" then? Is it the rank up to which the loss is evaluated as suggest in rows 23-24? Please clarify.

This is correct. “n” is now defined, thanks for pointing this out.

R3.15:

727-29: Did you test alternative strategies than averaging over days, like taking the maximum? Averaging might under-predict species with shorter blooming period, no?

In the context of identifying potentially dominant tree species we have not explored alternative ways of averaging over days. Preliminary exploration showed that observation probabilities for tree species were more stable over the year than for herbaceous species. Acutally, given that they are among the only plants identifiable in winter, their observation probability even increases in the cold season (however, we removed the coldest months before taking averages). It may well be that alternative ways of summarizing observation probabilities would have slightly improved our results, but in this analysis we did not have the capacity to thoroughly explore all settings applied to the various analyses. This is the exiting part of multispecies distribution modelling with DNNs, the applications seem endless and we have only started exploring them.

Bibliography

1. Scherrer, D., Esperon-Rodriguez, M., Beaumont, L. J., Barradas, V. L. & Guisan, A. National assessments of species vulnerability to climate change strongly depend on selected data sources. *Divers. Distrib.* **27**, 1367–1382 (2021).
2. Shen, X. & Meinshausen, N. Engression: Extrapolation for Nonlinear Regression? (2023).
3. Phillips, S. J. *et al.* Sample selection bias and presence-only distribution models: implications for background and pseudo-absence data. *Ecol. Appl.* **19**, 181–197 (2009).
4. Righetti, D., Vogt, M., Gruber, N., Psomas, A. & Zimmermann, N. E. Global pattern of phytoplankton diversity driven by temperature and environmental variability. *Sci. Adv.* **5**, eaau6253 (2019).
5. Descombes, P. *et al.* *Strategies for sampling pseudo-absences for species distribution models in complex mountainous terrain.* (2022) doi:10.1101/2022.03.24.485693.